# LightSAM: Parameter-Agnostic Sharpness-Aware Minimization

## Abstract

Sharpness-Aware Minimization (SAM) optimizer enhances the generalization ability of the machine learning model by exploring the flat minima landscape through weight perturbations. Despite its empirical success, SAM introduces an additional hyper-parameter, the perturbation radius, which causes the sensitivity of SAM to it. Moreover, it has been proved that the perturbation radius and learning rate of SAM are constrained by problem-dependent parameters to guarantee convergence. These limitations indicate the requirement of parameter-tuning in practical applications. In this paper, we propose the algorithm LightSAM which sets the perturbation radius and learning rate of SAM adaptively, thus extending the application scope of SAM. LightSAM employs three popular adaptive optimizers, including AdaGrad-Norm, AdaGrad and Adam, to replace the SGD optimizer for weight perturbation and model updating, reducing sensitivity to parameters. Theoretical results show that under weak assumptions, LightSAM could converge ideally with any choices of perturbation radius and learning rate, thus achieving parameter-agnostic. We conduct preliminary experiments on several deep learning tasks, which together with the theoretical findings validate the the effectiveness of LightSAM.

## 1 Introduction

Machine learning has achieved significant success across various application domains. As a critical component of machine learning, many optimization approaches are explored to train the model efficiently. However, most of the previous works only focus on minimizing the training loss, which would face the dilemma of over-fitting since the popular models are over-parameterized. Recently, there has been a raised attention on generalization ability since it represents the prediction ability on unseen data, thus very crucial for a model. Keskar et al. (2016); Neyshabur et al. (2017) study the relationship between the flatness of loss landscape and generalization ability, which consequently suggests finding flat minima that have low curvature in the neighbourhoods.

The above idea is formalized as a novel minimax problem, named Sharpness-Aware Minimization (Foret et al., 2020). The main difference from the original loss function is that Sharpness-Aware Minimization has a step that maximizes the loss function in the neighbourhood. This consideration of worst-case guarantees the low loss value in a region, thus making the loss landscape of minima flat and improving generalization ability, which results in the novel SAM optimizer: in each iteration, a weight perturbation is performed along the gradient direction with radius $\rho$, then the stochastic gradient on the perturbed weight is used in gradient descent with learning rate $\eta$ to update the model. SAM significantly improves the test performances of several deep networks (Foret et al., 2020).

The convergence rates of SAM and its variants have been extensively analyzed in existing works (Andriushchenko & Flammarion, 2022; Mi et al., 2022; Shin et al., 2023; Sun et al., 2024). However, these theoretical results require restrictions on two hyper-parameters of perturbation radius $\rho$ and learning rate $\eta$, either upper bounded or unequal relationship between them. These restrictions usually involve some problem-dependent constants, such as the Lipschitz constant, whose value could not be obtained a prior and hard to be estimated. In addition, though it is proved that the normalization in the perturbation step makes SAM less sensitive on $\rho$ (Dai et al., 2023), the empirical studies in the above works show that the sensitivity to the learning rate still exists and the adopted values are not stable. These shortcomings make it necessary to do parameter-tuning in empirical studies, which increases cost especially when training large-scale models. Thus, we raise a question that:

*Can we make SAM parameter-agnostic[1]?*

In fact, parameter-agnostic algorithms are thoroughly studied in online learning to avoid parameter-tuning (Orabona, 2014; Cutkosky & Boahen, 2017; Orabona & Tommasi, 2017). Recently, Defazio & Mishchenko (2023) suggest to use Adagrad-like step size to achieve learning-rate-agnostic. Wang et al. (2023b) and Wang et al. (2023a) prove the ideal convergence rate for adaptive optimizers. These motivate us to introduce adaptive learning rate into SAM to realize parameter-agnostic. Note that directly introducing adaptivity for both the perturbation radius and learning rate is technically non-trivial. This is due to that the terms need to be bounded would involve two gradients in one iteration, and the relationship between them is hard to establish since the randomnesses in one term could not be decoupled directly in the proof for adaptive methods.

In this paper, we study how to make the SAM optimizer parameter-agnostic. To achieve this goal, we propose an algorithm LightSAM. We provide three options for LightSAM, and in each option, we adopt one commonly used adaptive optimizer to perform weight perturbation and model update instead of SGD in vanilla SAM. As a consequence, both the weight perturbation and model update become adaptive during training. Specifically, we adopt the AdaGrad-Norm-type learning rate for LightSAM, named LightSAM-I, which uses a scaler-type adaptive learning rate for both the perturbation ascent step and gradient descent step $(\rho, \eta)$. In addition, we also consider the AdaGrad-type and Adam-type learning rate for LightSAM, named LightSAM-II and LightSAM-III respectively, which use coordinate-wise learning rates for two hyper-parameters $(\rho, \eta)$. Theoretically, we prove the $O(\ln T/\sqrt{T})$ convergence rate for LightSAM without any restrictions on perturbation radius and learning rate, thus achieving parameter-agnostic optimizers. Additionally, we only require nearly the weakest assumptions among related studies.

Our contributions can be summarized as follows:

- We propose an algorithm LightSAM for non-convex optimization. Compared to SAM, our algorithm could adopt AdaGrad-Norm, AdaGrad or Adam to implement the weight perturbation and model update steps. As a result, both the perturbation radius and learning rate become adaptive adjusted without requiring problem-dependent unknown parameters.

- The theoretical analysis indicates that LightSAM achieves the $O(\ln T/\sqrt{T})$ convergence rate without the gradient bounded assumption which is commonly used in adaptive optimizer analysis. Our result holds under any choices of hyper-parameters $(\rho, \eta)$, indicating that LightSAM is a parameter-agnostic optimizer, thereby saving the cost of parameter-tuning.

- We conduct several experiments to show the effectiveness of LightSAM, whose performance is stable under different parameter settings and coincides with our theoretical findings.

## 2 RELATED WORK

**Sharpness-Aware Minimization.** SAM optimizer (Foret et al., 2020) enhances the model generalization ability by minimizing the sharpness of loss landscape through an extra step of parameter perturbation. However, SAM still has some shortcomings in practical use, e.g., double gradient calculation and double learning rate hyper-parameter tuning. To address the issues where SAM exhibits insensitivity to parameter scaling, Kwon et al. (2021) propose ASAM. This method incorporates a normalization operator into the perturbation step to ensure adaptive sharpness. Recognizing the increased computational cost due to SAM's double forward and backward steps, SSAM (Mi et al., 2022) generates a mask to sparsify the perturbation while SAF (Du et al., 2022) replaces SAM's sharpness measure loss with a trajectory loss to achieve almost zero additional computation cost. GSAM (Zhuang et al., 2022) introduces an ascent step in the orthogonal direction to minimize the surrogate gap. Un-normalized SAM (USAM) (Andriushchenko & Flammarion, 2022) removes the normalization term in SAM and analyzes the convergence under standard assumptions. However, in order to guarantee the $O(1/\sqrt{T})$ convergence rate, the values of perturbation radius $\rho$ and learning rate $\eta$ are required to be dependent on the smoothness constant. Furthermore, Sun et al. (2024)

---

[1]In this paper, we follow the definition "parameter-agnostic" in Wang et al. (2024); Hübler et al. (2024) to describe an algorithm that could guarantee convergence with any parameter values. This implies that all parameters are not contingent upon any problem-dependent constants.

Table 1: Comparison between SAM-related works.

| Algorithm | Adaptive perturbation radius | Adaptive learning rate | Convergence rate[a] | Additional requirements |
|---|---|---|---|---|
| SAM | ✗ | ✗ | $O(\ln T/\sqrt{T})$[b] | Gradient bounded; Dependent on gradient bound |
| USAM | ✗ | ✗ | $O(1/\sqrt{T})$ | Dependent on Lipschitz constant |
| ASAM | ✓ | ✗ | - | - |
| AdaSAM | ✗ | ✓ | $O(1/\sqrt{T})$ | Dependent on Lipschitz constant; Gradient bounded |
| **LightSAM** | ✓ | ✓ | $O(\ln T/\sqrt{T})$ | None |

[a] "-" represents the convergence rate is not given in the original work.
[b] This could be improved to $O(1/\sqrt{T})$ by adjusting some hyper-parameters. We maintain the result in Mi et al. (2022).

propose the adaptive SAM by utilizing AMSGrad-type (Reddi et al., 2019) learning rate in SAM. However, the learning rate for maximizing the perturbation variable still requires heavy tuning.

**Adaptive Optimizer.** Adaptive optimizers make the learning rate adjust adaptively during the training process. Duchi et al. (2011) propose Adagrad, which accumulates the gradient second raw moment, i.e. the square of historical gradients, and makes the learning rate of each element inversely proportional to the square root of this sum. RMSProp (Tieleman, 2012) suggests adopting an exponential moving average for the stochastic gradients to make adaptive optimizer work well in deep learning. Adam (Kingma & Ba, 2014) further introduces the exponential moving average to the gradient second raw moment and becomes the most commonly used adaptive method.

It is showed that Adagrad could converge in both convex and non-convex settings (Li & Orabona, 2019). Adam-type algorithms achieve the $O(\ln T/\sqrt{T})$ convergence rate for non-convex optimization problems (Chen et al., 2018). The convergence rate $O(\sqrt{d/T})$ for AMSGrad, and $O(d/\sqrt{T})$ for Adagrad and RMSProp are theoretically proved (Zhou et al., 2018). Additionally, Défossez et al. (2020); Shen et al. (2023) analyze Adagrad and Adam under a framework with momentum and recover the $O(\ln T/\sqrt{T})$ convergence rate. However, most of these theoretical results rely on a strong assumption, i.e. the stochastic gradient is upper bounded. The analysis for RMSProp removes this assumption and concludes the convergence to a bounded region (Shi & Li, 2021). With the hyper-parameters commonly used in practice, Adam also converges to a region near critical points (Zhang et al., 2022). Recently, Wang et al. (2023b;a) make breakthroughs that recover the $O(\ln T/\sqrt{T})$ convergence rate without gradient bounded assumption.

**Parameter-Agnostic Optimization.** Parameter-agnostic (also known as parameter-free) algorithms are studied to achieve the optimal regret bound for the online optimization problem at first (Orabona, 2013; McMahan & Orabona, 2014; Orabona & Pál, 2016). Kernel-based SGD (Orabona, 2014) performs model selection and optimization without prior knowledge of problem and parameter-tuning. Orabona & Tommasi (2017) remove the learning rate from the gradient descent step to optimize the objective function. Carmon & Hinder (2022) focus on stochastic optimization and select the learning rate by a computable certificate. As a result, a nearly optimal convergence rate and parameter-agnostic are both achieved. D-Adaptation (Defazio & Mischenko, 2023) adopts Adagrad-like learning rate to iteratively lower bound the distance between the initial and optimal point . Normalized SGDM (Hübler et al., 2024) converges with a nearly optimal rate in the $(L_0, L_1)$-smoothness setting.

The above mentioned SAM-related works adopt SGD optimizer in weight perturbation or model update or both, which makes the parameters lack of adaptivity, and adaptive optimizer-related works seldom consider enhancing the generalization ability. Our work improves this by making both the perturbation radius and learning rate adaptive, and further parameter-agnostic. The most related work to this paper is Sun et al. (2024). However, it only employs the adaptive learning rate in the gradient descent step. Furthermore, their analysis requires the gradient bound assumption, which is too strong to be satisfied for practical applications (Nguyen et al., 2018). We also notice SA-SAM (Naganuma

et al.) which sets the learning rate by adaptively estimating the local smoothness constant, but it lacks of convergence guarantee. We list the comparison between these works and our work in Table 1.

## 3 METHODOLOGY

In this section, we propose a class of parameter-agnostic variants of SAM optimizer, named LightSAM. LightSAM could adopt the Adagrad-Norm-type learning rate (Levy, 2017; Ward et al., 2020), AdaGrad-type learning rate (Duchi et al., 2011) and Adam-type learning rate (Kingma & Ba, 2014) for estimating the double learning rate hyperparameters in SAM optimizer, denoted as LightSAM-I (AdaGrad-Norm), LightSAM-II (AdaGrad) and LightSAM-III (Adam) respectively. Below, we first introduce the problem setup for SAM and LightSAM.

### 3.1 PROBLEM SETUP

In this paper, we focus on the following stochastic non-convex optimization problem:

$$\min_{x \in R^d} f(x) := \frac{1}{n} \sum_{i=1}^{n} f(x, \xi_i),$$

where $f(x, \xi_i)$ denotes the loss function about model weights $x$ and data $\xi_i$, $n$ represents the number of training data. We further assume that this optimization problem is well-defined.

**Notations.** We use the following notations in this paper: $\|\cdot\|$ denotes the $l_2$ norm of a vector. $\nabla f(x)$ represents the gradient of function $f(x)$, $\nabla f(x)_l$ represents the $l$-th element of $\nabla f(x)$. For the vector sequences $\{a_t\}$, $a_{t,l}$ denotes the $l$-th element of $a_t$. $\odot$ represents element-wise multiplication.

**SAM Optimizer.** Sharpness-Aware Minimization problem (Foret et al., 2020) focuses on minimax saddle point optimization to seek a flat minimum by introducing the weight perturbation step

$$\min_x \max_{\|\epsilon\| \leq \rho} f_{\mathcal{S}}(x + \epsilon).$$

By alternatively performing a dual ascent step for the perturbation and a gradient descent step for the primal weight, SAM takes the following two-time scale update rule:

$$w_t = x_t + \rho \nabla f(x_t, \xi_t) / \|\nabla f(x_t, \xi_t)\|,$$
$$x_{t+1} = x_t - \eta \nabla f(w_t, \xi_t).$$

According to this update rule, SAM faces the challenge that there exist two learning rate hyperparameters $(\rho, \eta)$ that need to be carefully tuned. Dai et al. (2023) show that the learning rate $\rho$ for the perturbation step is crucial for the final performance of SAM. Classic trial-and-error learning tuning techniques for $\rho$ suffer from heavy tuning costs due to double gradient calculation in SAM. It is urgent to design cheap, lightweight, and automatic learning rate tuning techniques for SAM.

### 3.2 LIGHTSAM-I (ADAGRAD-NORM)

In this section, we propose our first algorithm LightSAM-I as described in Algorithm 1. Adagrad-Norm (Levy, 2017; Ward et al., 2020) only updates the scalar learning rate by historical gradients rather than the element-wise learning rate in AdaGrad. In the weight perturbation steps (lines 3-5) of our algorithm, we use the Adagrad-Norm to generate the perturbed weights $w_t$ instead of SGD optimizer in SAM. Meanwhile, we adopt the same strategy in the gradient descent steps (lines 6-8) to update model weights.

Before giving the theoretical analysis for Algorithm 1, we list some necessary assumptions. We

---

**Algorithm 1** LightSAM-I (AdaGrad-Norm)

**Require:** Initial values $x_0$, $u_0 = v_0 = \epsilon^2$, perturbation radius $\rho$, learning rate $\eta$.
1: **for** $t = 1, ..., T$ **do**
2:     Sample a minibatch $\xi_t$ from the dataset;
3:     Compute stochastic gradient $s_t = \nabla f(x_t, \xi_t)$;
4:     $u_t = u_{t-1} + \|s_t\|^2$;
5:     $w_t = x_t + \rho \frac{s_t}{\sqrt{u_t}}$;
6:     Compute stochastic gradient $g_t = \nabla f(w_t, \xi_t)$;
7:     $v_t = v_{t-1} + \|g_t\|^2$;
8:     Update weights $x_{t+1} = x_t - \eta \frac{g_t}{\sqrt{v_t}}$;
9: **end for**

---

denote $\mathcal{F}_t = \sigma\{s_1, g_1, ..., s_t, g_t\}$ as the sigma algebra generated by the observations of LightSAM after observing the stochastic gradients in the first $t$ iterations. $\mathbb{E}^{|\mathcal{F}_t}[\circ]$ is equivalent to $\mathbb{E}[\circ|\mathcal{F}_t]$.

**Assumption 1** (*L*-smoothness). *$f(x, \xi)$ is differentiable and satisfies the following inequality:*
$$\|\nabla f(x, \xi) - \nabla f(y, \xi)\| \leq L\|x - y\|, \forall x, y \in R^d.$$

**Assumption 2** (Affine noise variance). *There exist positive constants $(D_0, D_1)$ such that the following inequality holds:*
$$\mathbb{E}^{|\mathcal{F}_t}\|\nabla f(x_t, \xi_t)\|^2 \leq D_0 + D_1\|\nabla f(x_t)\|^2, \quad \mathbb{E}^{|\mathcal{F}_t}\|\nabla f(w_t, \xi_t)\|^2 \leq D_0 + D_1\|\nabla f(w_t)\|^2.$$

Straightforwardly, we could obtain the *L*-smoothness of $f(x)$ based on Assumption 1. These two assumptions are nearly the weakest requirements in stochastic optimization works, except that Assumption 1 assumes the *L*-smoothness of $f(x, \xi)$ instead of $f(x)$ as Assumption 1 in Wang et al. (2023b). This change is necessary in SAM-type works (Andriushchenko & Flammarion, 2022) since we need to establish the relationship between two stochastic gradients ($\nabla f(x_t, \xi_t)$ and $\nabla f(w_t, \xi_t)$) in one iteration.

**Technical Challenge.** In order to prove the convergence, we need to bound the term $\mathbb{E}\|\nabla f(x_t)\|^2$. However, LightSAM involves two stochastic gradients in one iteration. Thus when we want to bound the terms concerning $\mathbb{E}\|\nabla f(x_t)\|^2$, the upper bound would contain the terms concerning $\mathbb{E}\|\nabla f(w_t)\|^2$. On the other hand, the numerator and denominator of one term in adaptive optimization often share the same randomness which is hard to decouple. Thus, it is hard to analyze the inequality relationship between terms concerning $\mathbb{E}\|\nabla f(x_t)\|^2$ and $\mathbb{E}\|\nabla f(w_t)\|^2$.

By the above assumptions, we have the following theorem.

**Theorem 1.** *If $f(x)$ in Algorithm 1 satisfies Assumptions 1 and 2, for any perturbation radius $\rho$ and learning rate $\eta > 0$, we have that*
$$\frac{1}{T}\sum_{t=1}^{T}\mathbb{E}\|\nabla f(x_t)\|^2 \leq \frac{(2\sqrt{2D_0T + \epsilon^2} + A_5)(A_3 + 2A_4\ln(2\sqrt{2D_0T + \epsilon^2} + A_5))}{T}$$

*Here, we denote constants $D_2, A_1, A_2, A_3, A_4$ as following*
$$D_2 = \max\{1, D_1, \frac{8(1 + \sqrt{D_1})D_1\rho}{\eta}\}, \quad A_1 = \frac{\|\nabla f(w_1)\|^2}{\epsilon} + \frac{4(1 + 2D_2)L^2}{\epsilon}(\eta^2 - 2\rho^2\ln\epsilon),$$

$$A_2 = 2f(w_1) + 2\rho\|\nabla f(x_1)\| + 4\rho^2 L + \frac{D_0}{\epsilon}\eta + \frac{D_0\rho}{\epsilon\sqrt{D_1}} - (4L(1 + \rho L)(\eta^2 + 4\rho^2) + 2\rho)\ln\epsilon,$$

$$A_3 = \sqrt{\frac{\rho L}{\epsilon} + 1}[\frac{4D_0}{\epsilon} - (\frac{8\rho^2 L^2}{\epsilon} + 4\eta L)\ln\epsilon + \frac{4A_2}{\eta} + 8D_1A_1 + 8\eta L(2 + \rho L)\ln(1 + \frac{\rho L}{\epsilon})]$$

$$A_4 = \sqrt{\frac{\rho L}{\epsilon} + 1}[32\rho^2 L(1 + \rho L + \frac{(1 + 2D_2)D_1\eta L}{\epsilon} + \frac{\eta L}{8\epsilon}) + 4\rho + 8\eta^2 L(2 + \rho L)]/\eta,$$

$$A_5 = 4D_1A_3 + 4D_1A_4\ln(4D_1A_4).$$

**Corollary 1.** *From Theorem 1, we can obtain the following convergence rate for Algorithm 1*
$$\frac{1}{T}\sum_{t=1}^{T}\mathbb{E}\|\nabla f(x_t)\|^2 \leq O\left(\frac{\ln T}{\sqrt{T}}\right).$$

**Remark 1.** *Compared to previous works, the convergence rate of LightSAM recovers the result in works about adaptive optimizers (Zou et al., 2019; Défossez et al., 2020; Ward et al., 2020; Shi & Li, 2021; Shen et al., 2023; Wang et al., 2023b;a). When $T$ is sufficiently large, it converges with the same rate as USAM (Andriushchenko & Flammarion, 2022).*

**Remark 2.** *LightSAM not only requires nearly the lowest requirements on the assumptions but also has no restrictions on hyper-parameters, thus achieving parameter-agnostic.*

Due to limited space, we list the proof sketch here. The details could be referred to the Appendix.

*Proof Sketch.* The first part of our proof follows the proof of Wang et al. (2023b), i.e. our target is to bound $\sum_{t=1}^{T}\mathbb{E}\|\nabla f(x_t)\|^2/\sqrt{v_{t-1}}$. According to the smoothness of $f(x)$, we could obtain that

$$f(x_{T+1}) \leq f(x_1) + \eta\underbrace{\sum_{t=1}^{T}\langle\nabla f(x_t), \frac{-g_t}{\sqrt{v_{t-1}}}\rangle}_{T_1} + \eta\underbrace{\sum_{t=1}^{T}\langle\nabla f(x_t), \frac{g_t}{\sqrt{v_{t-1}}} - \frac{g_t}{\sqrt{v_t}}\rangle}_{T_2} + \frac{\eta^2 L}{2}\underbrace{\sum_{t=1}^{T}\|\frac{g_t}{\sqrt{v_t}}\|^2}_{T_3}$$

Since $T_1$ and $T_3$ is easy to bound:

$$\mathbb{E}[T_1] \leq -\frac{3\eta}{4}\sum_{t=1}^{T}\mathbb{E}\frac{\|\nabla f(x_t)\|^2}{\sqrt{v_{t-1}}} + \frac{\rho^2\eta L^2}{\epsilon}(\mathbb{E}\ln u_T - 2\ln\epsilon),$$

$$\mathbb{E}[T_3] \leq \frac{\eta^2 L}{2}(\mathbb{E}\ln v_T - \ln v_0) = \frac{\eta^2 L}{2}(\mathbb{E}\ln v_T - 2\ln\epsilon),$$

we turn to focus on $T_2$. Further, with appropriate scaling and Assumption 2, we obtain that

$$\mathbb{E}[T_2] \leq \frac{\eta}{4}\sum_{t=1}^{T}\mathbb{E}\frac{\|\nabla f(x_t)\|^2}{\sqrt{v_{t-1}}} + \frac{D_0\eta}{\epsilon} + D_1\eta\sum_{t=1}^{T}\|\nabla f(w_t)\|^2\mathbb{E}(\frac{1}{\sqrt{v_{t-1}}} - \frac{1}{\sqrt{v_t}}) \quad (1)$$

The last term in the similar proof step of Wang et al. (2023b) is $\sum_{t=1}^{T}\|\nabla f(x_t)\|^2\mathbb{E}(\frac{1}{\sqrt{v_{t-1}}} - \frac{1}{\sqrt{v_t}})$ which could be bounded by desired term $\sum_{t=1}^{T}\mathbb{E}\|\nabla f(x_t)\|^2/\sqrt{v_{t-1}}$. However, it does not apply to our proof since SAM-type algorithms involve different weights $x_t$ and $w_t$. Thus, it is non-trivial to bound the last term in (1). We give the following two lemmas to fill this gap.

**Lemma 1.** *If $f(x)$ in Algorithm 1 satisfies Assumptions 1 and 2, we have that*

$$\sum_{t=1}^{T}\|\nabla f(w_t)\|^2\mathbb{E}(\frac{1}{\sqrt{v_{t-1}}} - \frac{1}{\sqrt{v_t}}) \leq A_1 - \mathbb{E}\frac{\|\nabla f(w_T)\|^2}{\sqrt{v_T}} + \frac{1}{2D_2}\sum_{t=1}^{T}\mathbb{E}\frac{\|\nabla f(w_t)\|^2}{\sqrt{v_{t-1}}}$$

$$+ \frac{4(1+2D_2)\rho^2 L^2}{\epsilon}\mathbb{E}\ln u_T$$

**Lemma 2.** *If $f(x)$ in Algorithm 1 satisfies Assumptions 1 and 2, we have that*

$$\eta\sum_{t=1}^{T-1}\frac{\|\nabla f(w_t)\|^2}{\sqrt{v_{t-1}}} \leq 2\rho(1+\sqrt{D_1})\mathbb{E}\frac{\|\nabla f(x_T)\|^2}{\sqrt{u_{T-1}}} + D_1\eta\sum_{t=1}^{T-1}\mathbb{E}\|\nabla f(w_t)\|^2(\frac{1}{\sqrt{v_{t-1}}} - \frac{1}{\sqrt{v_t}})$$

$$+ A_2 + 2\eta^2 L(1+\rho L)\mathbb{E}\ln v_{T-1} + (8\rho^2 L(1+\rho L) + \rho)\mathbb{E}\ln u_T$$

Combining the above two lemmas and substituting the result into (1) helps us bound $T_2$ successfully. Then we establish the relationship between $v_t$ and $u_t$ as the following lemma:

**Lemma 3.** *If $f(x)$ in Algorithm 1 satisfies Assumption 1, we have that*

$$\|\nabla f(w_t, \xi_t)\|^2 \leq (\frac{\rho L}{\epsilon} + 1)\|\nabla f(x_t, \xi_t)\|^2, \quad v_t \leq (\frac{\rho L}{\epsilon} + 1)u_t$$

Up to this point, arranging the above results and substituting them into the first inequality yield that

$$\sum_{t=1}^{T}\mathbb{E}\frac{\|\nabla f(x_t)\|^2}{\sqrt{u_t}} \leq A_3 + A_4\mathbb{E}\ln u_T$$

Finally, we obtain that

$$\mathbb{E}\sqrt{u_T} \leq 2\sqrt{2D_0 T + \epsilon^2} + A_5$$

and the final result in Theorem 1 in the same way as Wang et al. (2023b). $\qquad\square$

**Discussion.** ASAM (Kwon et al., 2021) is proposed to alleviate the insensitivity of SAM to weight scaling . Though the element-wise operator is performed on the gradients to achieve sharpness adaptivity, the perturbation radius does not consider historical gradients like adaptive optimizers (Adagrad-Norm, Adagrad and Adam). AdaSAM (Sun et al., 2024) does not introduce adaptivity to the perturbation radius like LightSAM. Additionally, its theoretical analysis relies on a strong assumption, i.e. the stochastic gradient is upper bounded.

### 3.3 LIGHTSAM-II (ADAGRAD)

In LightSAM-II (see Algorithm 2), we adopt the AdaGrad-type learning rate to update the perturbation weights. LightSAM-II adopts the coordinate-wise learning rates to scale the perturbation step and gradient descent step, which can better utilize the historical gradients and achieve a stable convergence. Thus, compared to Algorithm 1, the initialized $u_0$ and $v_0$ become vectors with each element equal to $\epsilon^2$, and the multiplication and division become element-wise between vectors.

To provide the convergence of LightSAM-II with coordinate-wise learning rates, we require the following coordinate-wise smoothness and affine noise variance assumptions.

**Assumption 3** (Coordinate-wise $L$-smoothness). *For $\forall l \in [d]$, $f(x)$ is differentiable and satisfies:*

$$|\nabla f(x, \xi)_l - \nabla f(y, \xi)_l| \le L|x_l - y_l|, \forall x, y \in R^d.$$

**Assumption 4** (Coordinate-wise affine noise variance). *There exist positive constants $D_0$ and $D_1$:*

$$\nabla f(x, \xi)_l^2 \le D_0 + D_1 \nabla f(x)_l^2, \forall x \in R^d, \forall l \in [d].$$

---

**Algorithm 2** LightSAM-II (AdaGrad)

**Require:** Initial values $x_0$, $u_0 = v_0 = \epsilon^2$, perturbation radius $\rho$, learning rate $\eta$.
1: **for** $t = 1, ..., T$ **do**
2:     Sample a minibatch $\xi_t$ from the dataset;
3:     Compute stochastic gradient $s_t = \nabla f(x_t, \xi_t)$;
4:     $u_t = u_{t-1} + s_t \odot s_t$;
5:     $w_t = x_t + \rho \frac{1}{\sqrt{u_t}} \odot s_t$;
6:     Compute stochastic gradient $g_t = \nabla f(w_t, \xi_t)$;
7:     $v_t = v_{t-1} + g_t \odot g_t$;
8:     Update weights $x_{t+1} = x_t - \eta \frac{1}{\sqrt{v_t}} \odot g_t$;
9: **end for**

---

Assumption 3 is adopted in Richtárik & Takáč (2014); Das et al. (2024) and necessary here since the inequality relationship between $\nabla f(x_t, \xi_t)$ and $\nabla f(w_t, \xi_t)$ is established coordinate-wisely. Assumption 4 is commonly used in adaptive optimization works which do not need to assume the bounded gradient (Crawshaw et al., 2022; Wang et al., 2023b;a).

**Theorem 2.** *If $f(x)$ in Algorithm 2 satisfies Assumptions 3 and 4, for any perturbation radius $\rho$ and learning rate $\eta > 0$, we have that*

$$\frac{1}{T} \sum_{t=1}^{T} \mathbb{E}\|\nabla f(x_t)\|^2 \le \frac{(2\sqrt{2D_0 dT + \epsilon^2} + B_5)(B_3 + 2A_4 \ln(2\sqrt{2D_0 dT + \epsilon^2} + B_5))}{T}$$

*Here, we denote constants $B_1, B_2, B_3, B_5$ as following*

$$B_1 = \frac{\|\nabla f(w_1)\|^2}{\epsilon} + \frac{4(1 + 2D_2)dL^2}{\epsilon}(\eta^2 - 2\rho^2 \ln \epsilon),$$

$$B_2 = 2f(w_1) + d(2\rho\|\nabla f(x_1)\| + 4\rho^2 L + \frac{D_0 \eta}{\epsilon} + \frac{D_0 \rho}{\epsilon\sqrt{D_1}} - (4L(1 + \rho L)(\eta^2 + 4\rho^2) + 2\rho) \ln \epsilon)$$

$$B_3 = \sqrt{\frac{\rho L}{\epsilon} + 1}[\frac{4D_0 d}{\epsilon} - (\frac{8\rho^2 L^2}{\epsilon} + 4\eta L)d\ln \epsilon + 8D_1 B_1 + \frac{4B_2}{\eta}$$

$$+ \eta L(8(1 + \rho L) + 2)d\ln(1 + \frac{\rho L}{\epsilon})], \quad B_5 = 4D_1 B_3 + 4D_1 A_4 \ln(4D_1 A_4),$$

*and $D_2$ and $A_4$ are the same as Theorem 1.*

**Corollary 2.** *From Theorem 2, we can obtain the following convergence rate for Algorithm 2*

$$\frac{1}{T} \sum_{t=1}^{T} \mathbb{E}\|\nabla f(x_t)\|^2 \le O\left(\frac{\ln T}{\sqrt{T}}\right).$$

### 3.4 LIGHTSAM-III (ADAM)

Adam (Kingma & Ba, 2014) is another popular optimizer for deep learning, especially in Transformer-based models, which replaces the gradient aggregation step for estimating adaptive learning rate in AdaGrad with an exponential moving average step by introducing two additional momentum parameters $(\beta_1, \beta_2)$ and achieves a stable and fast convergence. In this section, we also integrate the Adam-type learning rate to update the parameters $(\rho, \eta)$ in SAM, which yields LightSAM-III (Adam), as shown in Algorithm 3. The convergence result for LightSAM-III is as follows:

---

**Algorithm 3** LightSAM-III (Adam)

**Require:** Initial values $x_0$, $r_0 = m_0 = 0$, $u_0 = v_0 = \epsilon^2$, perturbation radius $\rho$, learning rate $\eta$, coefficients $\beta_1, \beta_2$.
1: **for** $t = 1, ..., T$ **do**
2:     Sample a minibatch $\xi_t$ from the dataset;
3:     Compute stochastic gradient $s_t = \nabla f(x_t, \xi_t)$;
4:     $r_t = \beta_1 r_{t-1} + (1 - \beta_1)s_t$;
5:     $u_t = \beta_2 u_{t-1} + (1 - \beta_2)s_t \odot s_t$;
6:     $w_t = x_t + \rho \frac{1}{\sqrt{\epsilon^2 + u_t}} \odot r_t$;
7:     Compute stochastic gradient $g_t = \nabla f(w_t, \xi_t)$;
8:     $m_t = \beta_1 m_{t-1} + (1 - \beta_1)g_t$;
9:     $v_t = \beta_2 v_{t-1} + (1 - \beta_2)g_t \odot g_t$;
10:     Update weights $x_{t+1} = x_t - \eta \frac{1}{\sqrt{v_t}} \odot m_t$;
11: **end for**

---

Table 2: Best test accuracies (%) on MNIST dataset.

| Method | SGD | SAM | ASAM | AdaSAM | AdaGrad | L-SAM-II | Adam | L-SAM-III |
|--------|-----|-----|------|--------|---------|----------|------|-----------|
| 3-layer | 98.21 | 98.29 | 98.24 | 98.57 | 98.26 | 98.33 | 98.57 | **98.59** |
| LeNet | 99.29 | 99.37 | 99.48 | 99.48 | 99.25 | 99.31 | 99.41 | **99.49** |

Table 3: Average best test accuracies (%) of LightSAM under different hyper-parameters.

| Setting | 3-layer NN | | LeNet | |
|---------|------------|--|-------|--|
| | LightSAM-II | LightSAM-III | LightSAM-II | LightSAM-III |
| | 98.29±0.03 | 98.56±0.03 | 99.25 ± 0.07 | 99.41 ± 0.07 |

**Theorem 3.** *If $f(x)$ in Algorithm 3 satisfies Assumptions 3 and 4, and $0 \leq \beta_1 \leq \sqrt{\beta_2} - 32D_0(1 - \beta_2)/\beta^2$, $\beta_2 = 1 - \Theta(1/\sqrt{T})$. Then, for any perturbation radius $\rho = \Theta(1/\sqrt{T})$ and learning rate $\eta = \Theta(1/\sqrt{T})$, we have that*

$$\frac{1}{T}\sum_{t=1}^{T}\mathbb{E}\|\nabla f(x_t)\|^2 \leq O\left(\frac{\ln T}{\sqrt{T}}\right).$$

## 4 EXPERIMENTS

In this section, we conduct several experiments to show the effectiveness of our proposed algorithm. Experiments are conducted on MNIST and Imagenet datasets. The main goal of this paper is to validate that parameter-agnostic SAM optimizers without parameter tuning can achieve comparable performance with the carefully handcrafted learning rate schedule.

### 4.1 MNIST DATASET

**Implementation detail.** We first conduct the image classification task on the MNIST dataset. A simple 3-layer neural network and LeNet (LeCun et al., 1998) are adopted as the training models. We select SGD, AdaGrad, Adam, SAM, ASAM, AdaSAM, LightSAM-II and LightSAM-III as the baselines. The initial learning rate $\eta$ is set to 0.1 for SGD, SAM, and ASAM, 0.01 for AdaGrad and LightSAM-II, 0.001 for AdaSAM and LightSAM-III. The perturbation radius $\rho$ is set to 0.05 and 0.5 for SAM and ASAM respectively as suggested in Foret et al. (2020); Kwon et al. (2021), 0.1 for AdaSAM, 0.001 for LightSAM-II and III. We run all methods for 30 epochs. The learning rate is decayed two times by a factor of 0.2.

**Results on MNIST.** We summarize the best test accuracies of all baselines in the two experimental settings in Table 2. For each model, LightSAM-II achieves higher accuracy than AdaGrad, meanwhile, LightSAM-III achieves higher accuracy than Adam. This result indicates that parameter perturbation could improve the test accuracies of adaptive optimizers, the same as the phenomenon in the comparison between SAM and SGD. Additionally, LightSAM-II performs better than SAM in 3-layer neural network and LightSAM-III performs better than SAM in two cases, which is consistent with the advantage of Adam over SGD.

In the theoretical analysis, we prove that LightSAM could converge without tuning any hyper-parameters. Thus, in each experimental case, we scale the adopted $\rho$ and $\eta$ respectively, as a result obtaining four hyper-parameter settings $(\rho, 2\rho) * (\eta, 2\eta)$. We run LightSAM under these four settings and list the average result in Table 3. We can find that the average best accuracies are still higher than some baselines. The low standard deviations show the insensitivities of LightSAM to hyper-parameters.

### 4.2 FINETUNING ON IMAGENET DATASET

**Implementation detail.** We conduct the finetuning task on transformer models. Specifically, we finetune the ViT-Tiny and ViT-Small (Touvron et al., 2021) on the Imagenet-1k dataset for 10 epochs from the checkpoints pre-trained on the Imagenet-21k dataset. The utilized checkpoints are open-

Table 4: Best test accuracies (%) on Imagenet dataset after finetuning.

| Algorithms | SGD | Adam | SAM | ASAM | AdaSAM | LightSAM |
|---|---|---|---|---|---|---|
| ViT-Tiny | 45.59 | 60.82 | 60.10 | 59.95 | 64.43 | **64.58** |
| ViT-Small | 63.78 | 77.10 | 74.27 | 74.12 | 78.02 | **78.09** |

Table 5: Best test accuracies (%) of SAM-type algorithms under different parameter settings.

| SAM $(\eta,\rho)$=(0.1,0.05) | | | | | | | | | Avg. |
|---|---|---|---|---|---|---|---|---|---|
| 75.68 | 75.81 | 76.02 | 73.89 | 74.27 | 74.11 | 71.58 | 71.56 | 71.86 | $73.86 \pm 1.72$ |
| ASAM $(\eta,\rho)$=(0.1,0.5) | | | | | | | | | Avg. |
| 75.72 | 75.71 | 75.78 | 73.88 | 74.12 | 74.22 | 71.45 | -[a] | - | $74.41 \pm 1.44$ |
| AdaSAM $(\eta,\rho)$=(1e-4,0.01) | | | | | | | | | Avg. |
| 78.00 | 77.98 | 78.02 | 78.00 | 78.02 | 77.99 | 77.16 | 77.10 | 77.04 | $77.70 \pm 0.43$ |
| LightSAM $(\eta,\rho)$=(1e-4,1e-4) | | | | | | | | | Avg. |
| 77.97 | 78.00 | 78.04 | 77.99 | 78.09 | 78.06 | 77.29 | 77.10 | 77.27 | $\mathbf{77.76 \pm 0.38}$ |

[a] "-" represents the divergence of the algorithm.

sourced on `Huggingface`. We select SGD, Adam, SAM, ASAM, AdaSAM and LightSAM-III as the baselines. Following Foret et al. (2020); Kwon et al. (2021) and common choices, we set the learning rate as 0.1 for SGD, SAM and ASAM, 1e-4 for Adam, AdaSAM and LightSAM. And the perturbation radius is set as 0.05 for SAM, 0.5 for ASAM, 0.01 for AdaSAM and 1e-4 for LightSAM. Weight decay is not utilized for all optimizers. Momentum is set as 0.9 for all SGD optimizers.

**Results on Imagenet.** In Table 4, we list the best test accuracies of all baselines. Firstly, we could observe that the optimizers which adopt adaptive learning rate in the model update step (Adam, AdaSAM and LightSAM) perform better than those adopt constant learning rate (SGD, SAM and ASAM). This is in line with the advantage of adaptive optimizers over SGD on transformer based models (Zhang et al., 2020). Secondly, the optimizers utilize the weight perturbation step achieve higher test accuracies than the corresponding base optimizers (SAM and ASAM over SGD, AdaSAM and LightSAM over Adam), which presents the positive effect of weight perturbation in improving test performance. Finally, AdaSAM and LightSAM achieve comparable accuracies while LightSAM is still ahead of AdaSAM, thus the adaptive perturbation radius in LightSAM is comparable with the carefully handcrafted constant radius. We also show the illustration of the results in the Appendix.

**Sensitivity to hyper-parameters.** For several SAM-type algorithms, we enrich the experiment on a wide range of parameter values. For one baseline, denote the selected hyper-parameters in the above subsection as $\eta$ and $\rho$, we take nine combinations of parameters $(0.5\eta, \eta, 2\eta) * (0.5\rho, \rho, 2\rho)$ to show its sensitivity to these parameters. The results are shown in Table 5. The first nine columns record the best accuracy of one set of parameter values and the last column represents the mean and standard deviation. We could observe that SAM which does not have any adaptive modules has the highest deviation. ASAM does not converge in two settings with a large learning rate and performs worse than AdaSAM which adopts the commonly used adaptive learning rate. Under various parameter selections, our proposed algorithm achieves the highest mean accuracy and lowest deviation, which is in line with the "parameter-agnostic" property of LightSAM and indicates its insensitivity to the hyper-parameters.

### 4.3 FINETUNING ON GLUE TASK

**Implementation detail.** We also consider training the language models. We finetune the RoBERTa model (Liu, 2019) for 8 downstream tasks in the GLUE benchmark. The learning rate is set to 1e-2 for SGD, SAM and ASAM, 1e-5 for Adam, AdaSAM and LightSAM. The perturbation radius is set to 5e-3 for SAM and 1e-5 for LightSAM to maintain its ratio to learning rate same as the ViT experiment, 1e-2 for AdaSAM as adopted in (Sun et al., 2024), 1e-2 for ASAM after tuning. The batch size is set to 32 for all tasks except 16 for QNLI. We run all algorithms for 20 epochs.

**Results and parameter sensitivity on GLUE.** We list the experimental results in Table 6. We report the Matthew's correlation for CoLA, Pearson correlation for STS-B, F1 score for MRPC, averaged accuracy for MNLI, and accuracy for other tasks. Similar to the experiment on Imagenet,

Table 6: Experimental performances on GLUE benchmark after finetuning.

| Algorithms | CoLA | STS-B | MRPC | RTE | SST2 | MNLI | QNLI | QQP | Avg. |
|---|---|---|---|---|---|---|---|---|---|
| SGD | 59.39 | 87.85 | 91.65 | 76.53 | 93.69 | 86.33 | 89.27 | 91.49 | 84.53 |
| Adam | 62.08 | 90.77 | 92.50 | 78.70 | 94.84 | 87.42 | 92.82 | 91.90 | 86.38 |
| SAM | 61.71 | 89.25 | 92.01 | 79.42 | 94.27 | 86.42 | 89.53 | 91.38 | 85.50 |
| ASAM | 63.51 | 89.14 | 92.48 | 78.70 | 93.81 | 86.44 | 90.17 | 91.57 | 85.73 |
| AdaSAM | 62.11 | 90.55 | 93.12 | 80.14 | 95.30 | 87.57 | **93.10** | 92.01 | 86.74 |
| LightSAM | **63.77** | **90.77** | **93.33** | **81.95** | **95.41** | **87.63** | 92.92 | **92.04** | **87.23** |

Table 7: Performances of SAM-type algorithms under different parameter settings for STS-B.

| SAM $(\eta,\rho)$=(0.01,5e-3) | | | | | | | | | Avg. |
|---|---|---|---|---|---|---|---|---|---|
| - | 89.53 | 87.87 | 89.31 | 89.25 | 89.19 | - | - | - | $88.97 \pm 0.79$ |
| ASAM $(\eta,\rho)$=(0.01,0.01) | | | | | | | | | Avg. |
| 85.74 | 83.26 | - | 88.99 | 89.14 | 88.58 | - | - | - | $87.14 \pm 2.57$ |
| AdaSAM $(\eta,\rho)$=(1e-5,0.01) | | | | | | | | | Avg. |
| 90.20 | 90.29 | 90.27 | 90.54 | 90.55 | 90.48 | 90.86 | 91.01 | 90.92 | $90.57 \pm 0.30$ |
| LightSAM $(\eta,\rho)$=(1e-5,1e-5) | | | | | | | | | Avg. |
| 90.42 | 90.31 | 90.39 | 90.79 | 90.77 | 90.69 | 90.97 | 91.09 | 91.05 | $\mathbf{90.72 \pm 0.29}$ |

the algorithms that use the adaptive learning rate in the gradient descent step achieve the highest three scores, and each algorithm that adopts the perturbation step is ahead of its version that does not. Our proposed algorithm LightSAM performs best in seven tasks except the QNLI dataset, which again verifies its excellence in the practical application.

Samely, we conduct the experiments under nine sets of parameters $(0.5\eta, \eta, 2\eta) * (0.5\rho, \rho, 2\rho)$ on the STS-B task to test the sensitivity to the hyper-parameters for SAM-type optimizers, where $\eta$ and $\rho$ are the parameters set above. The results in Table 7 show the strong sensitivity of SAM and ASAM in this task as they fail to converge under four hyper-parameter settings. AdaSAM and LightSAM could converge to great solutions, which demonstrates the efficacy of the adaptive learning rate in the high stability. Between them, our proposed method has an advantage over AdaSAM, again indicating its insensitivity to the perturbation radius.

## 5 CONCLUSION

In this paper, we propose an algorithm LightSAM for non-convex optimization. LightSAM sets the perturbation radius and learning rate adaptively through adopting Adagrad-Norm, Adagrad, and Adam, respectively. We make a solid theoretical analysis for our proposed algorithm and observe that it converges with the $O(\ln T/\sqrt{T})$ rate without requiring the gradient bounded assumption. Particularly, our result does not require perturbation radius and learning rate satisfying any conditions, realizing parameter-agnostic optimizers. Finally, we conduct experiments in several computer vision tasks. The superiority of LightSAM to other baselines and the insensitivity to hyper-parameters are verified. Thus, we prove the potential of our work in reducing the necessity of parameter tuning from both theory and experiments.

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

## A    PROOF DETAILS

In this part, we show the proof details of theorems in the main body.

### A.1    USEFUL INEQUALITIES

We show some inequalities which are useful for our analysis.

**Lemma 4.** *(Lemma 10 in Wang et al. (2023b)) Consider sequence $\{a_t\}_{t=0}^T$ with $a_0 > 0, a_i \geq 0$ for $i > 0$, then we have*

$$\sum_{t=1}^T \frac{a_t}{\sum_{\tau=0}^t a_\tau} \leq \ln \sum_{t=0}^T a_t - \ln a_0,$$

$$\sum_{t=1}^T \frac{a_t}{(\sum_{\tau=0}^t a_\tau)^{3/2}} \leq \frac{2}{\sqrt{a_0}},$$

$$\sum_{t=1}^T \frac{a_t}{(\sum_{\tau=0}^t a_\tau)^{1/2}((\sum_{\tau=0}^{t-1} a_\tau)^{1/2} + (\sum_{\tau=0}^t a_\tau)^{1/2})^2} \leq \frac{1}{\sqrt{a_0}}.$$

**Lemma 5.** *(Lemmas 4 and 5 in Wang et al. (2023a)) Assume the constants $0 < \beta_1^2 < \beta_2 < 1$. Consider sequences $\{a_t\}_{t=1}^T$, $b_n = \beta_2 b_{n-1} + (1-\beta_2)a_n^2$ with $b_0 > 0$, $c_n = \beta_2 c_{n-1} + (1-\beta_2)a_n$ with $c_n = 0$, then we have*

$$\sum_{t=1}^T \frac{a_n^2}{b_n} \leq \frac{1}{1-\beta_2}(\ln \frac{b_T}{b_0} - T \ln \beta_2), \tag{2}$$

$$\sum_{t=1}^T \frac{c_n^2}{b_n} \leq \frac{(1-\beta_1)2}{(1-\frac{\beta_1}{\sqrt{\beta_2}})^2(1-\beta_2)}(\ln \frac{b_T}{b_0} - T \ln \beta_2). \tag{3}$$

### A.2    PROOF OF THEOREMS 1 AND 2

**Lemma 6.** *(Restatement of Lemma 1) If $f(x)$ in Algorithm 1 satisfies Assumptions 1 and 2, we have that*

$$\sum_{t=1}^T \mathbb{E}\|\nabla f(w_t)\|^2 (\frac{1}{\sqrt{v_{t-1}}} - \frac{1}{\sqrt{v_t}}) \leq A_1 - \mathbb{E}\frac{\|\nabla f(w_T)\|^2}{\sqrt{v_T}} + \frac{1}{2D_2}\sum_{t=1}^T \mathbb{E}\frac{\|\nabla f(w_t)\|^2}{\sqrt{v_{t-1}}}$$

$$+ \frac{4(1+2D_2)\rho^2 L^2}{\epsilon}\mathbb{E}\ln u_T \tag{4}$$

*where $D_2 = \max\{1, D_1, \frac{8(1+\sqrt{D_1})D_1\rho}{\eta}\}$, $A_1 = \frac{\|\nabla f(w_1)\|^2}{\epsilon} + \frac{4(1+2D_2)L^2}{\epsilon}(\eta^2 - 2\rho^2 \ln \epsilon)$.*

*Proof.* For two vectors $x$ and $y$, consider that $\langle x - y, y \rangle \leq \langle x - y, x \rangle$, we could further infer that $\langle x - y, y \rangle \leq \|x - y\|\|x\|$. And further $2\langle x, y \rangle - 2\|y\|^2 \leq 2\|x - y\|\|x\|$. Finally we obtain

$$\|x\|^2 - \|y\|^2 \leq 2\|x - y\|\|x\| + \|x\|^2 + \|y\|^2 - 2\langle x, y \rangle = 2\|x - y\|\|x\| + \|x - y\|^2$$

Based on this and Assumption 1, we have that

$$\mathbb{E}\|\nabla f(w_t)\|^2 (\frac{1}{\sqrt{v_{t-1}}} - \frac{1}{\sqrt{v_t}})$$

$$\leq \mathbb{E}[\frac{\|\nabla f(w_{t-1})\|^2}{\sqrt{v_{t-1}}} - \frac{\|\nabla f(w_t)\|^2}{\sqrt{v_t}}] + \frac{2L\|w_t - w_{t-1}\|\|\nabla f(w_t)\| + L^2\|w_t - w_{t-1}\|^2}{\sqrt{v_{t-1}}} \tag{5}$$

Consider

$$\|w_t - w_{t-1}\| \leq \eta\frac{\|\nabla f(w_{t-1}, \xi_{t-1})\|}{\sqrt{v_{t-1}}} + \rho\|\frac{\nabla f(x_t, \xi_t)}{\sqrt{u_t}} - \frac{\nabla f(x_{t-1}, \xi_{t-1})}{\sqrt{u_{t-1}}}\| \tag{6}$$

$$\|w_t - w_{t-1}\|^2 \leq 2\eta^2\frac{\|\nabla f(w_{t-1}, \xi_{t-1})\|^2}{v_{t-1}} + 2\rho^2\|\frac{\nabla f(x_t, \xi_t)}{\sqrt{u_t}} - \frac{\nabla f(x_{t-1}, \xi_{t-1})}{\sqrt{u_{t-1}}}\|^2 \tag{7}$$

Substituting (6) and (7) into (5) and summing the result over $t \in \{2, ..., T\}$ yields that

$$
\sum_{t=2}^{T} \mathbb{E}\|\nabla f(w_t)\|^2 (\frac{1}{\sqrt{v_{t-1}}} - \frac{1}{\sqrt{v_t}})
$$

$$
\leq \quad \mathbb{E}[\frac{\|\nabla f(w_1)\|^2}{\sqrt{v_1}} - \frac{\|\nabla f(w_T)\|^2}{\sqrt{v_T}}] + 2\eta L \sum_{t=2}^{T} \mathbb{E}\frac{\|\nabla f(w_{t-1}, \xi_{t-1})\|\|\nabla f(w_t)\|}{v_{t-1}}
$$

$$
+ 2\rho L \sum_{t=2}^{T} \mathbb{E}\frac{\|\frac{\nabla f(x_t, \xi_t)}{\sqrt{u_t}} - \frac{\nabla f(x_{t-1}, \xi_{t-1})}{\sqrt{u_{t-1}}}\|\|\nabla f(w_t)\|}{\sqrt{v_{t-1}}}
$$

$$
+ 2\eta^2 L^2 \sum_{t=2}^{T} \mathbb{E}\frac{\|\nabla f(w_{t-1}, \xi_{t-1})\|^2}{v_{t-1}^{3/2}} + 2\rho^2 L^2 \sum_{t=2}^{T} \mathbb{E}\frac{\|\frac{\nabla f(x_t, \xi_t)}{\sqrt{u_t}} - \frac{\nabla f(x_{t-1}, \xi_{t-1})}{\sqrt{u_{t-1}}}\|^2}{\sqrt{v_{t-1}}} \quad (8)
$$

In the RHS of (8)

$$
2\eta L \sum_{t=2}^{T} \mathbb{E}\frac{\|\nabla f(w_{t-1}, \xi_{t-1})\|\|\nabla f(w_t)\|}{v_{t-1}}
$$

$$
\leq \quad 4D_2\eta^2 L^2 \sum_{t=2}^{T} \mathbb{E}\frac{\|\nabla f(w_{t-1}, \xi_{t-1})\|^2}{v_{t-1}^{3/2}} + \frac{1}{4D_2} \sum_{t=2}^{T} \mathbb{E}\frac{\|\nabla f(w_t)\|^2}{\sqrt{v_{t-1}}}
$$

$$
2\rho L \sum_{t=2}^{T} \frac{\|\frac{\nabla f(x_t, \xi_t)}{\sqrt{u_t}} - \frac{\nabla f(x_{t-1}, \xi_{t-1})}{\sqrt{u_{t-1}}}\|\|\nabla f(w_t)\|}{\sqrt{v_{t-1}}}
$$

$$
\leq \quad 4D_2\rho^2 L^2 \sum_{t=2}^{T} \mathbb{E}\frac{\|\frac{\nabla f(x_t, \xi_t)}{\sqrt{u_t}} - \frac{\nabla f(x_{t-1}, \xi_{t-1})}{\sqrt{u_{t-1}}}\|^2}{\sqrt{v_{t-1}}} + \frac{1}{4D_2} \sum_{t=2}^{T} \mathbb{E}\frac{\|\nabla f(w_t)\|^2}{\sqrt{v_{t-1}}}
$$

Thus, we have

$$
\sum_{t=2}^{T} \mathbb{E}\|\nabla f(w_t)\|^2 (\frac{1}{\sqrt{v_{t-1}}} - \frac{1}{\sqrt{v_t}})
$$

$$
\leq \quad \mathbb{E}[\frac{\|\nabla f(w_1)\|^2}{\sqrt{v_1}} - \frac{\|\nabla f(w_T)\|^2}{\sqrt{v_T}}] + 2(1+2D_2)\eta^2 L^2 \sum_{t=2}^{T} \mathbb{E}\frac{\|\nabla f(w_{t-1}, \xi_{t-1})\|^2}{v_{t-1}^{3/2}}
$$

$$
+ \frac{1}{2D_2} \sum_{t=2}^{T} \mathbb{E}\frac{\|\nabla f(w_t)\|^2}{\sqrt{v_{t-1}}} + 2(1+2D_2)\rho^2 L^2 \sum_{t=2}^{T} \mathbb{E}\frac{\|\frac{\nabla f(x_t, \xi_t)}{\sqrt{u_t}} - \frac{\nabla f(x_{t-1}, \xi_{t-1})}{\sqrt{u_{t-1}}}\|^2}{\sqrt{v_{t-1}}}
$$

$$
\overset{(a)}{\leq} \quad \mathbb{E}[\frac{\|\nabla f(w_1)\|^2}{\sqrt{v_1}} - \frac{\|\nabla f(w_T)\|^2}{\sqrt{v_T}}] + \frac{1}{2D_2} \sum_{t=2}^{T} \mathbb{E}\frac{\|\nabla f(w_t)\|^2}{\sqrt{v_{t-1}}} + 4(1+2D_2)\eta^2 L^2\frac{1}{\epsilon}
$$

$$
+ \frac{4(1+2D_2)\rho^2 L^2}{\epsilon} \sum_{t=1}^{T} \mathbb{E}\frac{\|\nabla f(x_t, \xi_T)\|^2}{u_t}
$$

$$
\overset{(b)}{\leq} \quad \mathbb{E}[\frac{\|\nabla f(w_1)\|^2}{\sqrt{v_1}} - \frac{\|\nabla f(w_T)\|^2}{\sqrt{v_T}}] + \frac{1}{2D_2} \sum_{t=1}^{T} \mathbb{E}\frac{\|\nabla f(w_t)\|^2}{\sqrt{v_{t-1}}} + 4(1+2D_2)\eta^2 L^2\frac{1}{\epsilon}
$$

$$
+ \frac{4(1+2D_2)\rho^2 L^2}{\epsilon}(\mathbb{E}\ln u_T - \ln u_0)
$$

where (a) and (b) come from Lemma 4. Finally, we have

$$
\sum_{t=1}^{T} \mathbb{E}\|\nabla f(w_t)\|^2 \left(\frac{1}{\sqrt{v_{t-1}}} - \frac{1}{\sqrt{v_t}}\right)
$$

$$
\leq \quad \mathbb{E}\left[\frac{\|\nabla f(w_1)\|^2}{\sqrt{v_0}} - \frac{\|\nabla f(w_T)\|^2}{\sqrt{v_T}}\right] + \frac{1}{2D_2}\sum_{t=1}^{T}\mathbb{E}\frac{\|\nabla f(w_t)\|^2}{\sqrt{v_{t-1}}} + 4(1 + 2D_2)\eta^2 L^2 \frac{1}{\epsilon}
$$

$$
+ \frac{4(1 + 2D_2)\rho^2 L^2}{\epsilon}(\mathbb{E}\ln u_T - \ln u_0)
$$

$$
\leq \quad \frac{\|\nabla f(w_1)\|^2}{\epsilon} + \frac{4(1 + 2D_2)L^2}{\epsilon}(\eta^2 - 2\rho^2 \ln \epsilon) - \mathbb{E}\frac{\|\nabla f(w_T)\|^2}{\sqrt{v_T}}
$$

$$
+ \frac{1}{2D_2}\sum_{t=1}^{T}\mathbb{E}\frac{\|\nabla f(w_t)\|^2}{\sqrt{v_{t-1}}} + \frac{4(1 + 2D_2)\rho^2 L^2}{\epsilon}\mathbb{E}\ln u_T
$$

$\square$

**Lemma 7.** *(Restatement of Lemma 2) If $f(x)$ in Algorithm 1 satisfies Assumptions 1 and 2, we have that*

$$
\eta \sum_{t=1}^{T-1}\mathbb{E}\frac{\|\nabla f(w_t)\|^2}{\sqrt{v_{t-1}}} \leq D_1\eta\sum_{t=1}^{T-1}\mathbb{E}\|\nabla f(w_t)\|^2\left(\frac{1}{\sqrt{v_{t-1}}} - \frac{1}{\sqrt{v_t}}\right) + 2\rho(1 + \sqrt{D_1})\mathbb{E}\frac{\|\nabla f(x_T)\|^2}{\sqrt{u_{T-1}}}
$$

$$
+ A_2 + 2\eta^2 L(1 + \rho L)\mathbb{E}\ln v_{T-1} + (8\rho^2 L(1 + \rho L) + \rho)\mathbb{E}\ln u_T
$$

*where $A_2 = 2f(w_1) + 2\rho\|\nabla f(x_1)\| + 4\rho^2 L + \frac{D_0}{\epsilon}(\eta + \frac{\rho}{\sqrt{D_1}}) - (4L(1 + \rho L)(\eta^2 + 4\rho^2) + 2\rho)\ln \epsilon.$*

*Proof.* According to the $L$-smoothness of $f(x)$, we have

$$
\mathbb{E}^{|\mathcal{F}_t}[f(w_{t+1})] \leq \quad f(w_t) + \mathbb{E}^{|\mathcal{F}_t}\langle\nabla f(w_t), w_{t+1} - w_t\rangle + \frac{L}{2}\mathbb{E}^{|\mathcal{F}_t}\|w_{t+1} - w_t\|^2
$$

$$
= \quad f(w_t) + \eta\mathbb{E}^{|\mathcal{F}_t}\langle\nabla f(w_t), -\frac{\nabla f(w_t, \xi_t)}{\sqrt{v_t}}\rangle
$$

$$
+ \mathbb{E}^{|\mathcal{F}_t}\langle\nabla f(w_t), \rho(\frac{\nabla f(x_{t+1}, \xi_{t+1})}{\sqrt{u_{t+1}}} - \frac{\nabla f(x_t, \xi_t)}{\sqrt{u_t}})\rangle + \frac{L}{2}\mathbb{E}^{|\mathcal{F}_t}\|w_{t+1} - w_t\|^2
$$

$$(9)$$

Since

$$
\mathbb{E}^{|\mathcal{F}_t}\langle\nabla f(w_t), -\frac{\nabla f(w_t, \xi_t)}{\sqrt{v_t}}\rangle
$$

$$
= \quad -\mathbb{E}^{|\mathcal{F}_t}\langle\nabla f(w_t), \frac{\nabla f(w_t, \xi_t)}{\sqrt{v_{t-1}}}\rangle + \mathbb{E}^{|\mathcal{F}_t}\langle\nabla f(w_t), \nabla f(w_t, \xi_t)(\frac{1}{\sqrt{v_{t-1}}} - \frac{1}{\sqrt{v_t}})\rangle
$$

$$
= \quad -\frac{\|\nabla f(w_t)\|^2}{\sqrt{v_{t-1}}} + \mathbb{E}^{|\mathcal{F}_t}\langle\nabla f(w_t), \nabla f(w_t, \xi_t)(\frac{1}{\sqrt{v_{t-1}}} - \frac{1}{\sqrt{v_t}})\rangle \qquad (10)
$$

Substituting (10) into (9), we have

$$
\eta\frac{\|\nabla f(w_t)\|^2}{\sqrt{v_{t-1}}} \leq \quad f(w_t) - \mathbb{E}^{|\mathcal{F}_t}f(w_{t+1}) + \eta\mathbb{E}^{|\mathcal{F}_t}\langle\nabla f(w_t), \nabla f(w_t, \xi_t)(\frac{1}{\sqrt{v_{t-1}}} - \frac{1}{\sqrt{v_t}})\rangle
$$

$$
+ \mathbb{E}^{|\mathcal{F}_t}\langle\nabla f(w_t), \rho(\frac{\nabla f(x_{t+1}, \xi_{t+1})}{\sqrt{u_{t+1}}} - \frac{\nabla f(x_t, \xi_t)}{\sqrt{u_t}})\rangle
$$

$$
+ \frac{L}{2}\mathbb{E}^{|\mathcal{F}_t}\|w_{t+1} - w_t\|^2 \qquad (11)
$$

For the terms on the RHS of (11), first we have

$$
\mathbb{E}^{|\mathcal{F}_t} \|w_{t+1} - w_t\|^2
$$
$$
\leq \quad 2\eta^2 \mathbb{E}^{|\mathcal{F}_t} \frac{\|\nabla f(w_t, \xi_t)\|^2}{v_t} + 2\rho^2 \mathbb{E}^{|\mathcal{F}_t} \|\frac{\nabla f(x_{t+1}, \xi_{t+1})}{\sqrt{u_{t+1}}} - \frac{\nabla f(x_t, \xi_t)}{\sqrt{u_t}}\|^2
$$
$$
\leq \quad 2\eta^2 \mathbb{E}^{|\mathcal{F}_t} \frac{\|\nabla f(w_t, \xi_t)\|^2}{v_t} + 4\rho^2 \mathbb{E}^{|\mathcal{F}_t} (\frac{\|\nabla f(x_{t+1}, \xi_{t+1})\|^2}{u_{t+1}} + \frac{\|\nabla f(x_t, \xi_t)\|^2}{u_t}). \quad (12)
$$

Taking the expectation over $\mathcal{F}_t$ and summing up over $t \in \{1, 2, ..., T-1\}$ yields that

$$
\sum_{t=1}^{T-1} \mathbb{E}\|w_{t+1} - w_t\|^2 \leq 2\eta^2 (\mathbb{E}\ln v_{T-1} - \ln v_0) + 8\rho^2 (\mathbb{E}\ln u_T - \ln u_0). \quad (13)
$$

Then, we have

$$
\mathbb{E}^{|\mathcal{F}_t} \langle \nabla f(w_t), \nabla f(w_t, \xi_t)(\frac{1}{\sqrt{v_{t-1}}} - \frac{1}{\sqrt{v_t}})
$$
$$
\overset{(a)}{\leq} \quad \mathbb{E}^{|\mathcal{F}_t} \frac{\|\nabla f(w_t)\|\|\nabla f(w_t, \xi_t)\|^3}{\sqrt{v_{t-1}}\sqrt{v_t}(\sqrt{v_{t-1}} + \sqrt{v_t})} \overset{(b)}{\leq} \mathbb{E}^{|\mathcal{F}_t} \frac{\|\nabla f(w_t)\|\|\nabla f(w_t, \xi_t)\|^2}{\sqrt{v_{t-1}}(\sqrt{v_{t-1}} + \sqrt{v_t})}
$$
$$
\leq \quad \frac{1}{2}\mathbb{E}^{|\mathcal{F}_t} \frac{\|\nabla f(w_t)\|^2}{\sqrt{v_{t-1}}} + \frac{1}{2}(\mathbb{E}^{|\mathcal{F}_t} \frac{\|\nabla f(w_t, \xi_t)\|^2}{v_{t-1}^{1/4}(\sqrt{v_{t-1}} + \sqrt{v_t})})^2
$$
$$
\overset{(c)}{\leq} \quad \frac{1}{2}\mathbb{E}^{|\mathcal{F}_t} \frac{\|\nabla f(w_t)\|^2}{\sqrt{v_{t-1}}} + \frac{1}{2}\frac{1}{\sqrt{v_{t-1}}}(\mathbb{E}^{|\mathcal{F}_t}\|\nabla f(w_t, \xi_t)\|^2)(\mathbb{E}^{|\mathcal{F}_t} \frac{\|\nabla f(w_t, \xi_t)\|^2}{(\sqrt{v_{t-1}} + \sqrt{v_t})^2})
$$
$$
\overset{(d)}{\leq} \quad \frac{1}{2}\mathbb{E}^{|\mathcal{F}_t} \frac{\|\nabla f(w_t)\|^2}{\sqrt{v_{t-1}}} + \frac{1}{2}\sum_{t=1}^{T-1}\frac{1}{\sqrt{v_{t-1}}}(D_0 + D_1\|\nabla f(w_t)\|^2)(\mathbb{E}^{|\mathcal{F}_t} \frac{\|\nabla f(w_t, \xi_t)\|^2}{(\sqrt{v_{t-1}} + \sqrt{v_t})^2})
$$
$$
\overset{(e)}{\leq} \quad \frac{1}{2}\mathbb{E}^{|\mathcal{F}_t} \frac{\|\nabla f(w_t)\|^2}{\sqrt{v_{t-1}}} + \frac{D_0}{2}\mathbb{E}^{|\mathcal{F}_t} \frac{\|\nabla f(w_t, \xi_t)\|^2}{\sqrt{v_{t-1}}(\sqrt{v_{t-1}} + \sqrt{v_t})^2} + \frac{D_1}{2}\|\nabla f(w_t)\|^2 \mathbb{E}^{|\mathcal{F}_t}(\frac{1}{\sqrt{v_{t-1}}} - \frac{1}{\sqrt{v_t}})
$$
$$
(14)
$$

where (a) holds because of $\langle x, y \rangle \leq \|x\|\|y\|$; (b) holds because $\|\nabla f(w_t, \xi_t)\| \leq \sqrt{v_t}$; (c) comes from Cauchy's Inequality; (d) comes from Assumption 2; (e) holds because

$$
\frac{\|\nabla f(w_t, \xi_t)\|^2}{\sqrt{v_{t-1}}(\sqrt{v_{t-1}} + \sqrt{v_t})^2} \leq \frac{\|\nabla f(w_t, \xi_t)\|^2}{\sqrt{v_{t-1}}\sqrt{v_t}(\sqrt{v_{t-1}} + \sqrt{v_t})} = \frac{1}{\sqrt{v_{t-1}}} - \frac{1}{\sqrt{v_t}}, \quad (15)
$$

Taking the expectation on (14) over $\mathcal{F}_t$ and summing up over $t \in \{1, 2, ..., T-1\}$ yields that

$$
\sum_{t=1}^{T-1} \mathbb{E}\langle \nabla f(w_t), \nabla f(w_t, \xi_t)(\frac{1}{\sqrt{v_{t-1}}} - \frac{1}{\sqrt{v_t}})\rangle
$$
$$
\overset{(f)}{\leq} \quad \frac{1}{2}\sum_{t=1}^{T-1}\mathbb{E}\frac{\|\nabla f(w_t)\|^2}{\sqrt{v_{t-1}}} + \frac{D_0}{2\epsilon} + \frac{D_1}{2}\sum_{t=1}^{T-1}\mathbb{E}\|\nabla f(w_t)\|^2(\frac{1}{\sqrt{v_{t-1}}} - \frac{1}{\sqrt{v_t}}) \quad (16)
$$

where (f) comes from Lemma 4.

Finally, we have

$$\sum_{t=1}^{T-1} \mathbb{E}^{|\mathcal{F}_t} \langle \nabla f(w_t), \rho(\frac{\nabla f(x_{t+1}, \xi_{t+1})}{\sqrt{u_{t+1}}} - \frac{\nabla f(x_t, \xi_t)}{\sqrt{u_t}}) \rangle$$

$$= \rho \sum_{t=1}^{T-1} \mathbb{E}^{|\mathcal{F}_t} \langle \nabla f(w_{t+1}), \frac{\nabla f(x_{t+1}, \xi_{t+1})}{\sqrt{u_{t+1}}} \rangle - \mathbb{E}^{|\mathcal{F}_t} \langle \nabla f(w_t), \frac{\nabla f(x_t, \xi_t)}{\sqrt{u_t}} \rangle$$

$$+ \mathbb{E}^{|\mathcal{F}_t} \langle \nabla f(w_t) - \nabla f(w_{t+1}), \frac{\nabla f(x_{t+1}, \xi_{t+1})}{\sqrt{u_{t+1}}} \rangle$$

$$= \mathbb{E}^{|\mathcal{F}_T} \langle \nabla f(w_T), \rho \frac{\nabla f(x_T, \xi_T)}{\sqrt{u_T}} \rangle - \mathbb{E}^{|\mathcal{F}_1} \langle \nabla f(w_1), \rho \frac{\nabla f(x_1, \xi_1)}{\sqrt{u_1}} \rangle$$

$$+ \rho \sum_{t=1}^{T-1} \mathbb{E}^{|\mathcal{F}_t} \langle \nabla f(w_t) - \nabla f(w_{t+1}), \frac{\nabla f(x_{t+1}, \xi_{t+1})}{\sqrt{u_{t+1}}} \rangle \tag{17}$$

For the first term on the RHS of (17)

$$\mathbb{E}^{|\mathcal{F}_T} \langle \nabla f(w_T), \rho \frac{\nabla f(x_T, \xi_T)}{\sqrt{u_T}} \rangle$$

$$= \mathbb{E}^{|\mathcal{F}_T} \langle \nabla f(x_T + \rho \frac{\nabla f(x_T, \xi_T)}{\sqrt{u_T}}) - \nabla f(x_T), \rho \frac{\nabla f(x_T, \xi_T)}{\sqrt{u_T}} \rangle + \rho \mathbb{E}^{|\mathcal{F}_T} \langle \nabla f(x_T), \frac{\nabla f(x_T, \xi_T)}{\sqrt{u_T}} \rangle$$

$$\overset{(g)}{\le} \rho^2 L + \rho \mathbb{E}^{|\mathcal{F}_T} \langle \nabla f(x_T), \frac{\nabla f(x_T, \xi_T)}{\sqrt{u_{T-1}}} \rangle + \rho \mathbb{E}^{|\mathcal{F}_T} \langle \nabla f(x_T), \nabla f(x_T, \xi_T)(\frac{1}{\sqrt{u_T}} - \frac{1}{\sqrt{u_{T-1}}}) \rangle$$

$$\overset{(h)}{\le} \rho^2 L + \rho \frac{\|\nabla f(x_T)\|^2}{\sqrt{u_{T-1}}} + \rho \mathbb{E}^{|\mathcal{F}_T} \frac{\|\nabla f(x_T)\| \|\nabla f(x_T, \xi_T)\|^3}{\sqrt{u_{T-1}}\sqrt{u_T}(\sqrt{u_{T-1}} + \sqrt{u_T})}$$

$$\overset{(i)}{\le} \rho^2 L + \rho \mathbb{E}^{|\mathcal{F}_T} \frac{\|\nabla f(x_T)\|^2}{\sqrt{u_{T-1}}} + \rho \frac{\|\nabla f(x_T)\| \|\nabla f(x_T, \xi_T)\|^2}{\sqrt{u_{T-1}}(\sqrt{u_{T-1}} + \sqrt{u_T})} \tag{18}$$

where (g) holds because $\langle a, b \rangle \le \|a\| \|b\|$ and Assumption 1; (h) and (i) hold in the same way as (14).
For the last term on the RHS of (18)

$$\mathbb{E}^{|\mathcal{F}_T} \frac{\|\nabla f(x_T)\| \|\nabla f(x_T, \xi_T)\|^2}{\sqrt{u_{T-1}}(\sqrt{u_{T-1}} + \sqrt{u_T})}$$

$$\le \frac{\sqrt{D_1}}{2} \frac{\|\nabla f(x_T)\|^2}{\sqrt{u_{T-1}}} + \frac{1}{2\sqrt{D_1}\sqrt{u_{T-1}}} (\mathbb{E}^{|\mathcal{F}_T} \frac{\|\nabla f(x_T, \xi_T)\|^2}{\sqrt{u_{T-1}} + \sqrt{u_T}})^2$$

$$\le \frac{\sqrt{D_1}}{2} \frac{\|\nabla f(x_T)\|^2}{\sqrt{u_{T-1}}} + \frac{1}{2\sqrt{D_1}\sqrt{u_{T-1}}} (\mathbb{E}^{|\mathcal{F}_T} \|\nabla f(x_T, \xi_T)\|^2)(\mathbb{E}^{|\mathcal{F}_T} \frac{\|\nabla f(x_T, \xi_T)\|^2}{(\sqrt{u_{T-1}} + \sqrt{u_T})^2})$$

$$\le \frac{\sqrt{D_1}}{2} \frac{\|\nabla f(x_T)\|^2}{\sqrt{u_{T-1}}} + \frac{1}{2\sqrt{D_1}\sqrt{u_{T-1}}} (D_0 + D_1\|\nabla f(x_T)\|^2)(\mathbb{E}^{|\mathcal{F}_T} \frac{\|\nabla f(x_T, \xi_T)\|^2}{(\sqrt{u_{T-1}} + \sqrt{u_T})^2})$$

$$\overset{(j)}{\le} \sqrt{D_1} \frac{\|\nabla f(x_T)\|^2}{\sqrt{u_{T-1}}} + \frac{D_0}{2\epsilon\sqrt{D_1}} \tag{19}$$

where (j) holds because $\frac{\|\nabla f(x_T, \xi_T)\|^2}{(\sqrt{u_{T-1}} + \sqrt{u_T})^2} \le 1$ and $\sqrt{u_{T-1}} \ge \epsilon$. Combining (18) and (19) yields

$$\mathbb{E}^{|\mathcal{F}_T} \langle \nabla f(w_T), \rho \frac{\nabla f(x_T, \xi_T)}{\sqrt{u_T}} \rangle \le \rho^2 L + \frac{\rho D_0}{2\epsilon\sqrt{D_1}} + (1 + \sqrt{D_1})\rho \frac{\|\nabla f(x_T)\|^2}{\sqrt{u_{T-1}}} \tag{20}$$

For the second term on the RHS of (17)

$$-\mathbb{E}^{|\mathcal{F}_1}\langle\nabla f(w_1), \rho\frac{\nabla f(x_1, \xi_1)}{\sqrt{u_1}}\rangle$$

$$= -\mathbb{E}^{|\mathcal{F}_1}\langle\nabla f(x_1 + \rho\frac{\nabla f(x_1,\xi_1)}{\sqrt{u_1}}) - f(x_1), \rho\frac{\nabla f(x_1,\xi_1)}{\sqrt{u_1}}\rangle - \mathbb{E}^{|\mathcal{F}_1}\langle\nabla f(x_1), \rho\frac{\nabla f(x_1,\xi_1)}{\sqrt{u_1}}\rangle$$

$$\leq \rho^2 L + \mathbb{E}^{|\mathcal{F}_1}\|\nabla f(x_1)\|\|\rho\frac{\nabla f(x_1,\xi_1)}{\sqrt{u_1}}\|$$

$$\leq \rho^2 L + \rho\|\nabla f(x_1)\| \tag{21}$$

For the last term on the RHS of (17)

$$\sum_{t=1}^{T-1}\mathbb{E}^{|\mathcal{F}_t}\langle\nabla f(w_t) - f(w_{t+1}), \frac{\nabla f(x_{t+1},\xi_{t+1})}{\sqrt{u_{t+1}}}\rangle$$

$$\overset{(k)}{\leq} \frac{L^2}{2}\sum_{t=1}^{T-1}\mathbb{E}^{|\mathcal{F}_t}\|w_{t+1} - w_t\|^2 + \frac{1}{2}\sum_{t=1}^{T-1}\mathbb{E}^{|\mathcal{F}_t}\frac{\|\nabla f(x_{t+1},\xi_{t+1})\|^2}{u_{t+1}}$$

$$\overset{(l)}{\leq} \eta^2 L^2(\mathbb{E}^{|\mathcal{F}_{T-1}}\ln v_{T-1} - \ln v_0) + 4\rho^2 L^2(\mathbb{E}^{|\mathcal{F}_T}\ln u_T - \ln u_0) + \frac{1}{2}(\mathbb{E}^{|\mathcal{F}_T}\ln u_T - \ln u_0) \tag{22}$$

where (k) comes from Assumption 1; the (l) comes from Lemma 4. Substituting (20), (21) and (22) into (17) and taking the expectation over $\mathcal{F}_t$ yield that

$$\sum_{t=1}^{T-1}\mathbb{E}\langle\nabla f(w_t), \rho(\frac{\nabla f(x_{t+1},\xi_{t+1})}{\sqrt{u_{t+1}}} - \frac{\nabla f(x_t,\xi_t)}{\sqrt{u_t}})\rangle$$

$$\leq 2\rho^2 L + \frac{\rho D_0}{2\epsilon\sqrt{D_1}} + \rho\|\nabla f(x_1)\| - (\rho\eta^2 L^2 + 4\rho^3 L^2 + \frac{\rho}{2})\ln u_0$$

$$+ (1 + \sqrt{D_1})\rho\mathbb{E}\frac{\|\nabla f(x_T)\|^2}{\sqrt{u_{T-1}}} + \rho\eta^2 L^2\mathbb{E}\ln v_{T-1} + (4\rho^3 L^2 + \frac{\rho}{2})\mathbb{E}\ln u_T. \tag{23}$$

Substituting (13), (16) and (23) into (11) yields that

$$\eta\sum_{t=1}^{T-1}\frac{\|\nabla f(w_t)\|^2}{\sqrt{v_{t-1}}} \leq f(w_1) + \rho\|\nabla f(x_1)\| + 2\rho^2 L + \frac{D_0}{2\epsilon}(\eta + \frac{\rho}{\sqrt{D_1}}) + \frac{\eta}{2}\sum_{t=1}^{T-1}\frac{\|\nabla f(w_t)\|^2}{\sqrt{v_{t-1}}}$$

$$+ \frac{D_1\eta}{2}\sum_{t=1}^{T-1}\mathbb{E}\|\nabla f(w_t)\|^2(\frac{1}{\sqrt{v_{t-1}}} - \frac{1}{\sqrt{v_t}}) - ((1 + \rho L)(\eta^2 + 4\rho^2)L + \frac{\rho}{2})\ln u_0$$

$$+ \eta^2 L(1 + \rho L)\mathbb{E}\ln v_{T-1} + (4\rho^3 L^2 + 4\rho^2 L + \frac{\rho}{2})\mathbb{E}\ln u_T + (1 + \sqrt{D_1})\rho\mathbb{E}\frac{\|\nabla f(x_T)\|^2}{\sqrt{u_{T-1}}}.$$

Rearranging the result and considering that $\ln u_0 = 2\ln\epsilon$ yields the result. $\qquad\square$

**Lemma 8.** *(Restatement of Lemma 3) If $f(x)$ in Algorithm 1 satisfies Assumptions 1, we have that*

$$\|\nabla f(w_t, \xi_t)\|^2 \leq (\frac{\rho L}{\epsilon} + 1)\|\nabla f(x_t, \xi_t)\|^2, \quad v_t \leq (\frac{\rho L}{\epsilon} + 1)u_t$$

*Proof.*

$$\|\nabla f(w_t, \xi_t)\|^2$$

$$= \|\nabla f(w_t, \xi_t) - \nabla f(x_t, \xi_t)\|^2 + 2\langle\nabla f(w_t, \xi_t) - \nabla f(x_t, \xi_t), \nabla f(x_t, \xi_t)\rangle + \|\nabla f(x_t, \xi_t)\|^2$$

$$\leq L^2\|w_t - x_t\|^2 + 2L\|w_t - x_t\|\|\nabla f(x_t, \xi_t)\| + \|\nabla f(x_t, \xi_t)\|^2$$

$$= \rho^2 L^2\frac{\|\nabla f(x_t, \xi_t)\|^2}{u_t} + 2\rho L\frac{\|\nabla f(x_t, \xi_t)\|}{\sqrt{u_t}}\|\nabla f(x_t, \xi_t)\| + \|\nabla f(x_t, \xi_t)\|^2$$

$$= (\frac{\rho L}{\sqrt{u_t}} + 1)^2\|\nabla f(x_t, \xi_t)\|^2 \leq (\frac{\rho L}{\epsilon} + 1)^2\|\nabla f(x_t, \xi_t)\|^2$$

where the last inequality holds because $u_t \geq u_0 = \epsilon^2$. Further, we can obtain $v_t \leq (\frac{\rho L}{\epsilon} + 1)u_t$. $\quad\square$

**Theorem 4.** *(Restatement of Theorem 1) If $f(x)$ in Algorithm 1 satisfies Assumptions 1 and 2, for any perturbation radius $\rho$ and learning rate $eta > 0$, we have that*

$$\frac{1}{T}\sum_{t=1}^{T}\mathbb{E}\|\nabla f(x_t)\|^2 \leq \frac{(2\sqrt{2D_0 T + \epsilon^2} + A_5)(A_3 + 2A_4 \ln(2\sqrt{2D_0 T + \epsilon^2} + A_5))}{T}$$

*where*

$$A_3 = \sqrt{\frac{\rho L}{\epsilon} + 1}\Big[\frac{4D_0}{\epsilon} + 8D_1 A_1 + \frac{4A_2}{\eta} - \big(\frac{8\rho^2 L^2}{\epsilon} + 4\eta L\big)\ln\epsilon + 8\eta L(2 + \rho L)\ln(1 + \frac{\rho L}{\epsilon})\Big],$$

$$A_4 = \sqrt{\frac{\rho L}{\epsilon} + 1}\Big[32\rho^2 L(1 + \rho L + \frac{(1 + 2D_2)D_1 \eta L}{\epsilon} + \frac{\eta L}{8\epsilon}) + 4\rho + 8\eta^2 L(2 + \rho L)\Big]/\eta,$$

$$A_5 = 4D_1 A_3 + 4D_1 A_4 \ln(4D_1 A_4).$$

*Proof.* According to the $L$-smoothness of $f(x)$, we have

$$
\begin{aligned}
\mathbb{E}^{|\mathcal{F}_t}[f(x_{t+1})] &\leq f(x_t) + \mathbb{E}^{|\mathcal{F}_t}\langle\nabla f(x_t), x_{t+1} - x_t\rangle + \frac{L}{2}\mathbb{E}^{|\mathcal{F}_t}\|x_{t+1} - x_t\|^2 \\
&= f(x_t) - \eta\mathbb{E}^{|\mathcal{F}_t}\langle\nabla f(x_t), \frac{g_t}{\sqrt{v_t}}\rangle + \frac{\eta^2 L}{2}\mathbb{E}^{|\mathcal{F}_t}\|\frac{g_t}{\sqrt{v_t}}\|^2 \\
&= f(x_t) + \underbrace{\eta\mathbb{E}^{|\mathcal{F}_t}\langle\nabla f(x_t), \frac{-g_t}{\sqrt{v_{t-1}}}\rangle}_{T_1} + \underbrace{\eta\mathbb{E}^{|\mathcal{F}_t}\langle\nabla f(x_t), g_t(\frac{1}{\sqrt{v_{t-1}}} - \frac{1}{\sqrt{v_t}})\rangle}_{T_2} \\
&\quad + \underbrace{\frac{\eta^2 L}{2}\mathbb{E}^{|\mathcal{F}_t}\|\frac{g_t}{\sqrt{v_t}}\|^2}_{T_3}
\end{aligned}
$$

For $T_1$,

$$
\begin{aligned}
T_1 &= \eta\mathbb{E}^{|\mathcal{F}_t}\langle\nabla f(x_t), \frac{-\nabla f(w_t)}{\sqrt{v_{t-1}}}\rangle = \eta\mathbb{E}^{|\mathcal{F}_t}\langle\nabla f(x_t), \frac{-\nabla f(x_t + \rho\frac{\nabla f(x_t, \xi_t)}{\sqrt{u_t}})}{\sqrt{v_{t-1}}}\rangle \\
&= \frac{\eta}{\sqrt{v_{t-1}}}\mathbb{E}^{|\mathcal{F}_t}\Big(\langle\nabla f(x_t), \nabla f(x_t) - \nabla f(x_t + \rho\frac{\nabla f(x_t, \xi_t)}{\sqrt{u_t}})\rangle - \langle\nabla f(x_t), \nabla f(x_t)\rangle\Big) \\
&\leq \frac{\eta}{4}\mathbb{E}^{|\mathcal{F}_t}\frac{\|\nabla f(x_t)\|^2}{\sqrt{v_{t-1}}} + \frac{\eta}{\sqrt{v_{t-1}}}\mathbb{E}^{|\mathcal{F}_t}\|\nabla f(x_t) - \nabla f(x_t + \rho\frac{\nabla f(x_t, \xi_t)}{\sqrt{u_t}})\|^2 \\
&\quad - \eta\mathbb{E}^{|\mathcal{F}_t}\frac{\|\nabla f(x_t)\|^2}{\sqrt{v_{t-1}}} \\
&\overset{(a)}{\leq} -\frac{3\eta}{4}\mathbb{E}^{|\mathcal{F}_t}\frac{\|\nabla f(x_t)\|^2}{\sqrt{v_{t-1}}} + \frac{\eta}{\sqrt{v_{t-1}}}\mathbb{E}^{|\mathcal{F}_t}\frac{\rho^2 L^2\|\nabla f(x_t, \xi_t)\|^2}{u_t} \\
&\leq -\frac{3\eta}{4}\mathbb{E}^{|\mathcal{F}_t}\frac{\|\nabla f(x_t)\|^2}{\sqrt{v_{t-1}}} + \frac{\rho^2\eta L^2}{\sqrt{v_0}}\mathbb{E}^{|\mathcal{F}_t}\frac{\|\nabla f(x_t, \xi_t)\|^2}{u_t}
\end{aligned}
$$

where (a) comes from Assumption 1. Taking the expectation on the above inequality over $\mathcal{F}_t$ and summing up over $t \in \{1, 2, ..., T\}$ yields that

$$\sum_{t=1}^{T}\eta\mathbb{E}\langle\nabla f(x_t), \frac{-g_t}{\sqrt{v_{t-1}}}\rangle \overset{(b)}{\leq} -\frac{3\eta}{4}\mathbb{E}\frac{\|\nabla f(x_t)\|^2}{\sqrt{v_{t-1}}} + \frac{\rho^2\eta L^2}{\epsilon}(\mathbb{E}\ln u_T - 2\ln\epsilon), \quad\quad (24)$$

where (b) comes from Lemma 4.

For $T_2$

$$
\begin{aligned}
T_2 &= \eta \mathbb{E}^{|\mathcal{F}_t} \langle \nabla f(x_t), \frac{\nabla f(w_t, \xi_t) \|\nabla f(w_t, \xi_t)\|^2}{\sqrt{v_{t-1}}\sqrt{v_t}(\sqrt{v_{t-1}} + \sqrt{v_t})} \rangle \leq \eta \mathbb{E}^{|\mathcal{F}_t} \frac{\|\nabla f(x_t)\|\|\nabla f(w_t, \xi_t)\|^3}{\sqrt{v_{t-1}}\sqrt{v_t}(\sqrt{v_{t-1}} + \sqrt{v_t})} \\
&\leq \eta \mathbb{E}^{|\mathcal{F}_t} \frac{\|\nabla f(x_t)\|\|\nabla f(w_t, \xi_t)\|^2}{\sqrt{v_{t-1}}(\sqrt{v_{t-1}} + \sqrt{v_t})} \\
&\leq \frac{\eta}{4} \mathbb{E}^{|\mathcal{F}_t} \frac{\|\nabla f(x_t)\|^2}{\sqrt{v_{t-1}}} + \frac{\eta}{\sqrt{v_{t-1}}} \left( \mathbb{E}^{|\mathcal{F}_t} \frac{\|\nabla f(w_t, \xi_t)\|^2}{\sqrt{v_{t-1}} + \sqrt{v_t}} \right)^2 \\
&\leq \frac{\eta}{4} \mathbb{E}^{|\mathcal{F}_t} \frac{\|\nabla f(x_t)\|^2}{\sqrt{v_{t-1}}} + \eta (\mathbb{E}^{|\mathcal{F}_t} \|\nabla f(w_t, \xi_t)\|^2) \left( \mathbb{E}^{|\mathcal{F}_t} \frac{\|\nabla f(w_t, \xi_t)\|^2}{\sqrt{v_{t-1}}(\sqrt{v_{t-1}} + \sqrt{v_t})^2} \right) \\
&\leq \frac{\eta}{4} \mathbb{E}^{|\mathcal{F}_t} \frac{\|\nabla f(x_t)\|^2}{\sqrt{v_{t-1}}} + D_0 \eta \sum_{t=1}^{T} \mathbb{E}^{|\mathcal{F}_t} \frac{\|\nabla f(w_t, \xi_t)\|^2}{\sqrt{v_{t-1}}(\sqrt{v_{t-1}} + \sqrt{v_t})^2} \\
&\quad + D_1 \eta \|\nabla f(w_t)\|^2 \mathbb{E}^{|\mathcal{F}_t} (\frac{1}{\sqrt{v_{t-1}}} - \frac{1}{\sqrt{v_t}})
\end{aligned}
$$

Taking the expectation on the above inequality over $\mathcal{F}_t$ and summing up over $t \in \{1, 2, ..., T\}$ yields that

$$
\begin{aligned}
&\sum_{t=1}^{T} \eta \mathbb{E} \langle \nabla f(x_t), g_t (\frac{1}{\sqrt{v_{t-1}}} - \frac{1}{\sqrt{v_t}}) \rangle \\
&\leq \frac{\eta}{4} \sum_{t=1}^{T} \mathbb{E} \frac{\|\nabla f(x_t)\|^2}{\sqrt{v_{t-1}}} + \frac{D_0 \eta}{\epsilon} + D_1 \eta \sum_{t=1}^{T} \|\nabla f(w_t)\|^2 \mathbb{E} (\frac{1}{\sqrt{v_{t-1}}} - \frac{1}{\sqrt{v_t}}) \quad (25)
\end{aligned}
$$

the proof of (25) follows the same way as (14). From Lemma 6, we can obtain that

$$
\begin{aligned}
D_1 \eta \sum_{t=1}^{T} \|\nabla f(w_t)\|^2 \mathbb{E} (\frac{1}{\sqrt{v_{t-1}}} - \frac{1}{\sqrt{v_t}}) &\leq \frac{4(1 + 2D_2) D_1 \rho^2 \eta L^2}{\epsilon} \mathbb{E} \ln u_T - D_1 \eta \mathbb{E} \frac{\|\nabla f(w_T)\|^2}{\sqrt{v_T}} \\
&\quad + D_1 \eta A_1 + \frac{D_1}{2D_2} \eta \sum_{t=1}^{T} \mathbb{E} \frac{\|\nabla f(w_t)\|^2}{\sqrt{v_{t-1}}} \quad (26)
\end{aligned}
$$

By Lemma 7, we can further obtain that

$$
\begin{aligned}
&- D_1 \eta \mathbb{E} \frac{\|\nabla f(w_T)\|^2}{\sqrt{v_T}} + \frac{D_1}{2D_2} \eta \sum_{t=1}^{T} \mathbb{E} \frac{\|\nabla f(w_t)\|^2}{\sqrt{v_{t-1}}} \\
&\leq -\frac{D_1}{2D_2} \eta \mathbb{E} \frac{\|\nabla f(w_T)\|^2}{\sqrt{v_T}} + \frac{D_1}{2D_2} \eta (\sum_{t=1}^{T-1} \mathbb{E} \frac{\|\nabla f(w_t)\|^2}{\sqrt{v_{t-1}}} + \frac{\|\nabla f(w_T)\|^2}{\sqrt{v_{T-1}}}) \\
&\leq \frac{D_1^2}{2D_2} \eta \sum_{t=1}^{T-1} \mathbb{E} \|\nabla f(w_t)\|^2 (\frac{1}{\sqrt{v_{t-1}}} - \frac{1}{\sqrt{v_t}}) + \frac{D_1}{2D_2} \eta \mathbb{E} \|\nabla f(w_T)\|^2 (\frac{1}{\sqrt{v_{T-1}}} - \frac{1}{\sqrt{v_T}}) \\
&\quad + \frac{A_2}{2} + \frac{\eta}{8} \mathbb{E} \frac{\|\nabla f(x_T)\|^2}{\sqrt{u_{T-1}}} + \eta^2 L(1 + \rho L) \mathbb{E} \ln v_{T-1} + (4\rho^2 L(1 + \rho L) + \frac{\rho}{2}) \mathbb{E} \ln u_T \\
&\leq \frac{D_1}{2} \eta \sum_{t=1}^{T} \mathbb{E} \|\nabla f(w_t)\|^2 (\frac{1}{\sqrt{v_{t-1}}} - \frac{1}{\sqrt{v_t}}) + \frac{A_2}{2} + \frac{\eta}{8} \mathbb{E} \frac{\|\nabla f(x_T)\|^2}{\sqrt{u_{T-1}}} \\
&\quad + \eta^2 L(1 + \rho L) \mathbb{E} \ln v_{T-1} + (4\rho^2 L(1 + \rho L) + \frac{\rho}{2}) \mathbb{E} \ln u_T \quad (27)
\end{aligned}
$$

Substituting (27) into (26) yields that

$$
\begin{aligned}
&D_1 \eta \sum_{t=1}^{T} \|\nabla f(w_t)\|^2 \mathbb{E} (\frac{1}{\sqrt{v_{t-1}}} - \frac{1}{\sqrt{v_t}}) \leq 2D_1 \eta A_1 + A_2 + \frac{\eta}{4} \mathbb{E} \frac{\|\nabla f(x_T)\|^2}{\sqrt{u_{T-1}}} \\
&\quad + 2\eta^2 L(1 + \rho L) \mathbb{E} \ln v_{T-1} + (8\rho^2 L(1 + \rho L + \frac{(1 + 2D_2) D_1 \eta L}{\epsilon}) + \rho) \mathbb{E} \ln u_T \quad (28)
\end{aligned}
$$

Substituting (28) into (25) yields that

$$
\sum_{t=1}^{T} \mathbb{E}\eta \mathbb{E}^{|\mathcal{F}_t} \langle \nabla f(x_t), g_t(\frac{1}{\sqrt{v_{t-1}}} - \frac{1}{\sqrt{v_t}}) \rangle \leq \frac{\eta}{4} \sum_{t=1}^{T} \mathbb{E}\frac{\|\nabla f(x_t)\|^2}{\sqrt{v_{t-1}}} + \frac{D_0 \eta}{\epsilon} + 2D_1 \eta A_1 + A_2
$$

$$
+\frac{\eta}{4}\mathbb{E}\frac{\|\nabla f(x_T)\|^2}{\sqrt{u_{T-1}}} + 2\eta^2 L(1 + \rho L)\mathbb{E}\ln v_{T-1} + (8\rho^2 L(1 + \rho L + \frac{(1 + 2D_2)D_1 \eta L}{\epsilon}) + \rho)\mathbb{E}\ln u_T
$$

$$(29)$$

For $T_3$, taking expectation over $\mathcal{F}_t$ and summing up over $t \in \{1, 2, ..., T\}$ yields that

$$
\frac{\eta^2 L}{2} \sum_{t=1}^{T} \mathbb{E}\|\frac{g_t}{\sqrt{v_t}}\|^2 \leq \frac{\eta^2 L}{2}(\mathbb{E}\ln v_T - \ln v_0) = \frac{\eta^2 L}{2}(\mathbb{E}\ln v_T - 2\ln \epsilon) \tag{30}
$$

Combining (24), (29) and (30) yields that

$$
\frac{\eta}{2} \sum_{t=1}^{T} \mathbb{E}\frac{\|\nabla f(x_t)\|^2}{\sqrt{v_{t-1}}}
$$

$$
\leq \quad \frac{D_0 \eta}{\epsilon} + 2D_1 \eta A_1 + A_2 - (\frac{2\rho^2 \eta L^2}{\epsilon} + \eta^2 L)\ln \epsilon + \eta^2 L(2(1 + \rho L) + \frac{1}{2})\mathbb{E}\ln v_T
$$

$$
+(8\rho^2 L(1 + \rho L + \frac{(1 + 2D_2)D_1 \eta L}{\epsilon} + \frac{\eta L}{8\epsilon}) + \rho)\mathbb{E}\ln u_T + \frac{\eta}{4}\mathbb{E}\frac{\|\nabla f(x_T)\|^2}{\sqrt{u_{T-1}}}
$$

$$
\leq \quad \frac{D_0 \eta}{\epsilon} + 2D_1 \eta A_1 + A_2 - (\frac{2\rho^2 \eta L^2}{\epsilon} + \eta^2 L)\ln \epsilon + \eta^2 L(2(1 + \rho L) + \frac{1}{2})\ln(1 + \frac{\rho L}{\epsilon})
$$

$$
+(8\rho^2 L(1 + \rho L + \frac{(1 + 2D_2)D_1 \eta L}{\epsilon} + \frac{\eta L}{8\epsilon}) + \rho + \eta^2 L(2(1 + \rho L) + \frac{1}{2}))\mathbb{E}\ln u_T
$$

$$
+\frac{\eta}{4} \sum_{t=1}^{T} \mathbb{E}\frac{\|\nabla f(x_T)\|^2}{\sqrt{u_{T-1}}} \tag{31}
$$

Rearranging the result and considering that $\frac{\|\nabla f(x_t)\|^2}{\sqrt{v_{t-1}}} \geq \sqrt{\frac{\epsilon}{\rho L + \epsilon}} \frac{\|\nabla f(x_t)\|^2}{\sqrt{u_{t-1}}}$ (which comes from Lemma 8) yields that

$$
\sum_{t=1}^{T} \mathbb{E}\frac{\|\nabla f(x_t)\|^2}{\sqrt{u_t}} \leq A_3 + A_4 \mathbb{E}\ln u_T
$$

Finally, we adopt the same derivation as "Stage II" in the proof of Lemma 4 in Wang et al. (2023b) to obtain that

$$
\mathbb{E}[\sqrt{u_T}] \leq 2\sqrt{2D_0 T + \epsilon^2} + A_5 \tag{32}
$$

as well as the final result.

$\square$

The proof of Theorem 2 is similar to the above proof. The difference is the scalars are replaced with vectors, and for vectors $a$ and $b$, we turn to bound $\|a \odot b\|^2 = \sum_{l=1}^{d} a_l^2 b_l^2$ and $\|\frac{1}{b} \odot a\|^2 = \sum_{l=1}^{d} \frac{a_l^2}{b_l^2}$. We do not repeat the whole progress here.

### A.3 PROOF OF THEOREM 3

Before the proof, we define

$$
p_t = \frac{w_t - \frac{\beta_1}{\sqrt{\beta_2}} w_{t-1}}{1 - \frac{\beta_1}{\sqrt{\beta_2}}},
$$

$$
\tilde{u}_t = \beta_2 u_{t-1} + (1 - \beta_2)D_0 \mathbb{1}_d,
$$

$$
q_t = \frac{x_t - \frac{\beta_1}{\sqrt{\beta_2}} x_{t-1}}{1 - \frac{\beta_1}{\sqrt{\beta_2}}},
$$

$$\tilde{v}_t = \beta_2 v_{-1} + (1 - \beta_2) D_0 \mathbb{1}_d.$$

The idea of proof of Theorem 3 is identical to that of Theorem 1. We provide the main intermediate results and omit some details.

**Lemma 9.** *If $f(x)$ in Algorithm 3 satisfies Assumptions 3 and 4, we have that*

$$
\frac{3}{4} \sum_{t=1}^{T} \sum_{l=1}^{d} \mathbb{E} \frac{\nabla f(w_t)_l^2}{\sqrt{\tilde{v}_{t,l}}} \leq C_0 \sum_{t=1}^{T} \sum_{l=1}^{d} \mathbb{E}\left(\frac{\nabla f(w_{t-1})_l^2}{\sqrt{\beta_2 \tilde{v}_{t,l}}} - \frac{\nabla f(w_t)_l^2}{\sqrt{v_{t,l}}}\right) + C_1 \sum_{l=1}^{d} \mathbb{E} \ln r_{t,l}^2
$$

$$
+ C_2 \sum_{l=1}^{d} \mathbb{E} \ln m_{t,l}^2 + C_3
$$

*Proof.* From the definition, we have that

$$
p_{t+1,i} - p_{t,i}
$$
$$
= -\frac{(1-\beta_1)\eta}{1-\frac{\beta_1}{\sqrt{\beta_2}}} \frac{g_{t,i}}{\sqrt{\tilde{v}_{t,i}}} - \frac{\eta}{1-\frac{\beta_1}{\sqrt{\beta_2}}}\left(\frac{1}{\sqrt{v_{t,i}}} - \frac{1}{\sqrt{\tilde{v}_{t,i}}}\right)m_{t,i} + \frac{\beta_1\eta}{1-\frac{\beta_1}{\sqrt{\beta_2}}}\left(\frac{1}{\sqrt{\beta_2 v_{t-1,i}}} - \frac{1}{\sqrt{\tilde{v}_{t,i}}}\right)m_{t-1,i}
$$
$$
-\frac{(1-\beta_1)\rho}{1-\frac{\beta_1}{\sqrt{\beta_2}}} \frac{s_{t,i}}{\sqrt{\tilde{u}_{t,i}}} - \frac{\rho}{1-\frac{\beta_1}{\sqrt{\beta_2}}}\left(\frac{1}{\sqrt{u_{t,i}}} - \frac{1}{\sqrt{\tilde{u}_{t,i}}}\right)r_{t,i} + \frac{\beta_1\rho}{1-\frac{\beta_1}{\sqrt{\beta_2}}}\left(\frac{1}{\sqrt{\beta_2 u_{t-1,i}}} - \frac{1}{\sqrt{\tilde{u}_{t,i}}}\right)r_{t-1,i}
$$
$$
-\frac{(1-\beta_1)\rho}{1-\frac{\beta_1}{\sqrt{\beta_2}}} \frac{s_{t-1,i}}{\sqrt{\tilde{u}_{t-1,i}}} - \frac{\rho}{1-\frac{\beta_1}{\sqrt{\beta_2}}}\left(\frac{1}{\sqrt{u_{t-1,i}}} - \frac{1}{\sqrt{\tilde{u}_{t-1,i}}}\right)r_{t-1,i}
$$
$$
+\frac{\beta_1\rho}{1-\frac{\beta_1}{\sqrt{\beta_2}}}\left(\frac{1}{\sqrt{\beta_2 u_{t-2,i}}} - \frac{1}{\sqrt{\tilde{u}_{t-1,i}}}\right)r_{t-2,i}
$$

According to the $L$-smoothness, we have that

$$
f(p_{t+1}) \leq f(p_t) + \langle \nabla f(w_t), p_{t+1} - p_t \rangle + \langle \nabla f(p_t) - \nabla f(w_t), p_{t+1} - p_t \rangle + \frac{L}{2}\|p_{t+1} - p_t\|^2 \quad (33)
$$

Summing up the above inequality over $\{1, ..., T\}$ and take the expectation yields that

$$
\mathbb{E}[f(p_{T+1})] \leq f(p_1) + \sum_{t=1}^{T} \mathbb{E}\langle \nabla f(w_t), p_{t+1} - p_t \rangle + \sum_{t=1}^{T} \mathbb{E}\langle \nabla f(p_t) - \nabla f(w_t), p_{t+1} - p_t \rangle
$$
$$
+ \frac{L}{2}\mathbb{E}\|p_{t+1} - p_t\|^2 \quad (34)
$$

For the term $\sum_{t=1}^{T} \mathbb{E}\langle \nabla f(w_t), p_{t+1} - p_t \rangle$, we follow Wang et al. (2023a) and the analysis in Lemma 7, bound each term as

$$
-\sum_{t=1}^{T} \sum_{l=1}^{d} \frac{g_{t,l}}{\sqrt{\tilde{v}_{t,l}}} \nabla f(w_t)_l \leq -\frac{3}{4} \sum_{t=1}^{T} \sum_{l=1}^{d} \mathbb{E} \frac{\nabla f(w_t)_l^2}{\sqrt{\tilde{v}_{t,l}}} + \rho^2 L^2 \sum_{t=1}^{T} \sum_{l=1}^{d} \mathbb{E} \frac{\frac{g_{t,l}^2}{v_{t,l}}}{\sqrt{\tilde{u}_{t,l}}} \quad (35)
$$

$$
-\sum_{t=1}^{T} \sum_{l=1}^{d} \frac{s_{t,l}}{\sqrt{\tilde{u}_{t,l}}} \nabla f(w_t)_l \leq -\frac{3}{4} \sum_{t=1}^{T} \sum_{l=1}^{d} \mathbb{E} \frac{\nabla f(x_t)_l^2}{\sqrt{\tilde{u}_{t,l}}} + \rho^2 L^2 \sum_{t=1}^{T} \sum_{l=1}^{d} \mathbb{E} \frac{\frac{r_{t,l}^2}{u_{t,l}}}{\sqrt{\tilde{u}_{t,l}}} \quad (36)
$$

$$
-\sum_{t=1}^{T} \sum_{l=1}^{d} \frac{s_{t-1,l}}{\sqrt{\tilde{u}_{t-1,l}}} \nabla f(w_t)_l \leq -\frac{3}{4} \sum_{t=1}^{T} \sum_{l=1}^{d} \mathbb{E} \frac{\nabla f(x_{t-1})_l^2}{\sqrt{\tilde{u}_{t-1,l}}} + \rho^2 L^2 \sum_{t=1}^{T} \sum_{l=1}^{d} \mathbb{E} \frac{\frac{r_{t,l}^2}{u_{t,l}}}{\sqrt{\tilde{u}_{t-1,l}}} \quad (37)
$$

$$
\sum_{t=1}^{T} \sum_{l=1}^{d}\left(\frac{1}{\sqrt{\beta_2 v_{t-1,l}}} - \frac{1}{\sqrt{\tilde{v}_{t,l}}}\right)m_{t-1,l} \nabla f(w_t)_l
$$
$$
\leq \frac{1}{16} \sum_{t=1}^{T} \sum_{l=1}^{d} \mathbb{E} \frac{\nabla f(w_t)_l^2}{\sqrt{\tilde{v}_{t,l}}} + \frac{4\beta_1 \sqrt{(1-\beta_2)D_0}}{(1-\beta_1)\beta_2} \sum_{t=1}^{T} \sum_{l=1}^{d} \mathbb{E} \frac{m_{t-1,l}^2}{v_{t,l}} \quad (38)
$$

$$\sum_{t=1}^{T}\sum_{l=1}^{d}(\frac{1}{\sqrt{\beta_2 u_{t-1,l}}} - \frac{1}{\sqrt{\tilde{u}_{t,l}}})r_{t-1,l}\nabla f(w_t)_l$$

$$\leq \quad \frac{1}{16}\sum_{t=1}^{T}\sum_{l=1}^{d}\mathbb{E}\frac{\nabla f(w_t)_l^2}{\sqrt{\tilde{u}_{t,l}}} + \frac{4\beta_1\sqrt{(1-\beta_2)D_0}}{(1-\beta_1)\beta_2}\sum_{t=1}^{T}\sum_{l=1}^{d}\mathbb{E}\frac{r_{t-1,l}^2}{u_{t,l}} \tag{39}$$

$$\sum_{t=1}^{T}\sum_{l=1}^{d}(\frac{1}{\sqrt{\beta_2 u_{t-2,l}}} - \frac{1}{\sqrt{\tilde{u}_{t-1,l}}})r_{t-2,l}\nabla f(w_t)_l$$

$$\leq \quad \frac{1}{16}\sum_{t=1}^{T}\sum_{l=1}^{d}\mathbb{E}\frac{\nabla f(w_t)_l^2}{\sqrt{\tilde{u}_{t-1,l}}} + \frac{4\beta_1\sqrt{(1-\beta_2)D_0}}{(1-\beta_1)\beta_2}\sum_{t=1}^{T}\sum_{l=1}^{d}\mathbb{E}\frac{r_{t-2,l}^2}{u_{t-1,l}} \tag{40}$$

$$\sum_{t=1}^{T}\sum_{l=1}^{d}(\frac{1}{\sqrt{\tilde{v}_{t,l}}} - \frac{1}{\sqrt{v_{t,l}}})m_{t,l}\nabla f(w_t)_l$$

$$\leq \quad \frac{(1-\beta_1)\eta}{2(1-\frac{\beta_1}{\sqrt{\beta_2}})}\sum_{t=1}^{T}\sum_{l=1}^{d}\mathbb{E}\frac{\nabla f(w_t)_l^2}{\sqrt{\tilde{v}_{t,l}}} + \frac{2\eta\sqrt{(1-\beta_2)D_0}}{(1-\frac{\beta_1^2}{\beta_2})^2}\sum_{t=1}^{T}\sum_{l=1}^{d}\mathbb{E}\frac{g_{t,l}^2}{v_{t,l}}$$

$$+\frac{4(1-\beta_1)\eta D_1}{(1-\frac{\beta_1}{\sqrt{\beta_2}})^2}\sum_{t=1}^{T}\sum_{l=1}^{d}\mathbb{E}(\frac{\nabla f(w_{t-1})_l^2}{\sqrt{\beta_2 \tilde{v}_{t,l}}} - \frac{\nabla f(w_t)_l^2}{\sqrt{v_{t,l}}})$$

$$+\frac{64(1-\beta_1)D_1(1+D_1)\eta L^2}{(1-\beta_1)\beta_2^2(1-\frac{\beta_1}{\sqrt{\beta_2}})^3\sqrt{(1-\beta_2)D_0}}\sum_{t=1}^{T}\sum_{l=1}^{d}\mathbb{E}(2\eta^2\frac{m_{t,l}^2}{v_t} + 8\rho^2\frac{r_{t,l}^2}{u_{t,l}})$$

$$+\frac{(1-\beta_1)\eta}{8(1-\frac{\beta_1}{\sqrt{\beta_2}})}\sum_{t=1}^{T}\sum_{l=1}^{d}\mathbb{E}\frac{\nabla f(w_t)_l^2}{\sqrt{\tilde{v}_{t,l}}} + \frac{2\eta\sqrt{(1-\beta_2)D_0}}{(1-\beta_1)(1-\frac{\beta_1}{\sqrt{\beta_2}})}\sum_{t=1}^{T}\sum_{l=1}^{d}\mathbb{E}\frac{m_{t,l}^2}{v_{t,l}^2} \tag{41}$$

$$\sum_{t=1}^{T}\sum_{l=1}^{d}(\frac{1}{\sqrt{\tilde{u}_{t,l}}} - \frac{1}{\sqrt{u_{t,l}}})r_{t,l}\nabla f(w_t)_l$$

$$\leq \quad \frac{(1-\beta_1)\eta}{2(1-\frac{\beta_1}{\sqrt{\beta_2}})}\sum_{t=1}^{T}\sum_{l=1}^{d}\mathbb{E}\frac{\nabla f(w_t)_l^2}{\sqrt{\tilde{u}_{t,l}}} + \frac{2\eta\sqrt{(1-\beta_2)D_0}}{(1-\frac{\beta_1^2}{\beta_2})^2}\sum_{t=1}^{T}\sum_{l=1}^{d}\mathbb{E}\frac{r_{t,l}^2}{u_{t,l}}$$

$$+\frac{4(1-\beta_1)\eta D_1}{(1-\frac{\beta_1}{\sqrt{\beta_2}})^2}\sum_{t=1}^{T}\sum_{l=1}^{d}\mathbb{E}(\frac{\nabla f(w_{t-1})_l^2}{\sqrt{\beta_2 \tilde{u}_{t,l}}} - \frac{\nabla f(w_t)_l^2}{\sqrt{u_{t,l}}})$$

$$+\frac{64(1-\beta_1)D_1(1+D_1)\eta L^2}{(1-\beta_1)\beta_2^2(1-\frac{\beta_1}{\sqrt{\beta_2}})^3\sqrt{(1-\beta_2)D_0}}\sum_{t=1}^{T}\sum_{l=1}^{d}\mathbb{E}(2\eta^2\frac{m_{t,l}^2}{v_t} + 8\rho^2\frac{r_{t,l}^2}{u_{t,l}})$$

$$+\frac{(1-\beta_1)\eta}{8(1-\frac{\beta_1}{\sqrt{\beta_2}})}\sum_{t=1}^{T}\sum_{l=1}^{d}\mathbb{E}\frac{\nabla f(w_t)_l^2}{\sqrt{\tilde{u}_{t,l}}} + \frac{2\eta\sqrt{(1-\beta_2)D_0}}{(1-\beta_1)(1-\frac{\beta_1}{\sqrt{\beta_2}})}\sum_{t=1}^{T}\sum_{l=1}^{d}\mathbb{E}\frac{r_{t,l}^2}{u_{t,l}^2} \tag{42}$$

For the other two terms, we have

$$\sum_{t=1}^{T}\mathbb{E}\langle\nabla f(p_t) - \nabla f(w_t), p_{t+1} - p_t\rangle + \frac{L}{2}\mathbb{E}\|p_{t+1} - p_t\|^2$$

$$\leq \quad \eta^2 L\left(2\left(\frac{\frac{\beta_1}{\sqrt{\beta_2}}}{1-\frac{\beta_1}{\sqrt{\beta_2}}}\right)^2\mathbb{E}\|\frac{1}{\sqrt{v_{t-1}}}\odot m_{t-1}\|^2 + 2\left(\frac{1}{1-\frac{\beta_1}{\sqrt{\beta_2}}}\right)^2\mathbb{E}\|\frac{1}{\sqrt{v_t}}\odot m_t\|^2\right)$$

$$+\rho^2 L\left(4\left(\frac{\frac{\beta_1}{\sqrt{\beta_2}}}{1-\frac{\beta_1}{\sqrt{\beta_2}}}\right)^2\mathbb{E}\|\frac{1}{\sqrt{u_{t-1}}}\odot r_{t-1}\|^2 + 3\left(\frac{1}{1-\frac{\beta_1}{\sqrt{\beta_2}}}\right)^2\mathbb{E}\|\frac{1}{\sqrt{u_t}}\odot r_t\|^2\right) \tag{43}$$

Summing up the above results yields that

$$
\begin{aligned}
\frac{3}{4} \sum_{t=1}^{T} \sum_{l=1}^{d} \mathbb{E} \frac{\nabla f(w_t)_l^2}{\sqrt{\tilde{v}_{t,l}}} \le \ & C_0 \sum_{t=1}^{T} \sum_{l=1}^{d} \mathbb{E}\left( \frac{\nabla f(w_{t-1})_l^2}{\sqrt{\beta_2 \tilde{v}_{t,l}}} - \frac{\nabla f(w_t)_l^2}{\sqrt{v_{t,l}}} \right) \\
& + C_1 \sum_{l=1}^{d} \mathbb{E} \ln r_{t,l}^2 + C_2 \sum_{l=1}^{d} \mathbb{E} \ln m_{t,l}^2 + C_3
\end{aligned}
\tag{44}
$$

where $C_0$, $C_1$, $C_2$ and $C_3$ are constants with respect to $\eta$, $\rho$, $\beta_1$, $\beta_2$, $D_0$ and $D_1$. $\qquad\square$

**Lemma 10.** *If $f(x)$ in Algorithm 3 satisfies Assumptions 3, we have that*

$$
\tilde{v}_{t,l} \le C \tilde{u}_{t,l},
\tag{45}
$$

*where the constant $C = \max\{1, 2(1-\beta_2)[1 + \frac{(1-\beta_1)^2 \rho^2 L^2}{(1-\beta_1^a)(1-\beta_2^b)\epsilon^2}]\}$.*

*Proof.*

$$
\begin{aligned}
g_{t,l}^2 &= \left( \nabla f(x_t + \rho \frac{r_t}{\sqrt{u_t + \epsilon^2}}, \xi_t)_l - \nabla f(x_t, \xi_t)_l + \nabla f(x_t, \xi_t)_l \right)^2 \\
&\le \frac{2\rho^2 L^2}{\epsilon^2} r_{t,l}^2 + 2 s_{t,l}^2 \\
&= \frac{2(1-\beta_1)^2 \rho^2 L^2}{\epsilon^2} \sum_{\tau=1}^{t} (\beta_1^{t-\tau} s_{\tau,l})^2 + 2 s_{t,l}^2.
\end{aligned}
\tag{46}
$$

Thus, we have that

$$
\begin{aligned}
v_{t,l} &= (1-\beta_2) \sum_{k=1}^{t} \beta_2^{t-k} g_{k,l}^2 + \beta_2^t \epsilon^2 \\
&\le \frac{2(1-\beta_1)^2(1-\beta_2)\rho^2 L^2}{\epsilon^2} \sum_{k=1}^{t} \beta_2^{t-k} \sum_{\tau=1}^{k} (\beta_1^{k-\tau} s_{\tau,l})^2 + 2(1-\beta_2) \sum_{k=1}^{t} \beta_2^{t-k} s_{k,l}^2 + \beta_2^t \epsilon^2.
\end{aligned}
\tag{47}
$$

Since $\beta_1 < \sqrt{\beta_2}$, there exists constants $0 < a, b < 2$ satisfy that $\beta_1^{2-a} \le \beta_2^{1+b}$. Then, we have that

$$
\begin{aligned}
\sum_{k=1}^{t} \beta_2^{t-k} \sum_{\tau=1}^{k} (\beta_1^{k-\tau} s_{\tau,l})^2 &\le \sum_{k=1}^{t} \beta_2^{t-k} \left( \sum_{\tau=1}^{k} \beta_1^{a(k-\tau)} \right) \left( \sum_{\tau=1}^{k} \beta_1^{(2-a)(k-\tau)} s_{\tau,l}^2 \right) \\
&\le \frac{1}{1-\beta_1^a} \sum_{k=1}^{t} \beta_2^{t-k} \sum_{\tau=1}^{k} \beta_1^{(2-a)(k-\tau)} s_{\tau,l}^2 \\
&= \frac{1}{1-\beta_1^a} \sum_{k=1}^{t} \left( \sum_{j=0}^{t-k} \beta_1^{(2-a)j} \beta_2^{t-k-j} \right) s_{k,l}^2 \\
&\le \frac{1}{1-\beta_1^a} \sum_{k=1}^{t} \beta_2^{t-k} \left( \sum_{j=0}^{t-k} \beta_2^{bj} \right) s_{k,l}^2 \\
&\le \frac{1}{(1-\beta_1^a)(1-\beta_2^b)} \sum_{k=1}^{t} \beta_2^{t-k} s_{k,l}^2.
\end{aligned}
\tag{48}
$$

Substituting (48) into (47) yields that

$$
\begin{aligned}
v_{t,l} &\le 2(1-\beta_2) \left[ 1 + \frac{(1-\beta_1)^2 \rho^2 L^2}{(1-\beta_1^a)(1-\beta_2^b)\epsilon^2} \right] \sum_{k=1}^{t} \beta_2^{t-k} s_{k,l}^2 + \beta_2^t \epsilon^2 \\
&\le C u_{t,l}.
\end{aligned}
\tag{49}
$$

Finally, considering the definition of $\tilde{v}_{t,l}$, we have that

$$
\tilde{v}_{t,l} \le C u_{t,l}.
\tag{50}
$$

$\qquad\square$

**Theorem 5.** *(Restatement of Theorem 3) If $f(x)$ in Algorithm 3 satisfies Assumptions 3 and 4, and $0 \le \beta_1 \le \sqrt{\beta_2} - 32D_0(1 - \beta_2)/\beta^2$, $\beta_2 = 1 - \Theta(1/\sqrt{T})$. Then, for any perturbation radius $\rho = \Theta(1/\sqrt{T})$ and learning rate $\eta = \Theta(1/\sqrt{T})$, we have that*

$$\frac{1}{T}\sum_{t=1}^{T}\mathbb{E}\|\nabla f(x_t)\|^2 \le O\left(\frac{\ln T}{\sqrt{T}}\right).$$

*Proof.* From the definition, we have that

$$q_{t+1,i} - q_{t,i} = -\eta\frac{1-\beta_1}{1-\frac{\beta_1}{\sqrt{\beta_2}}}\frac{g_{t,i}}{\sqrt{\tilde{v}_{t,i}}} - \eta\frac{1}{1-\frac{\beta_1}{\sqrt{\beta_2}}}\left(\frac{1}{\sqrt{v_{t,i}}} - \frac{1}{\sqrt{\tilde{v}_{t,i}}}\right)m_{t,i}$$

$$+ \eta\frac{\beta_1}{1-\frac{\beta_1}{\sqrt{\beta_2}}}\left(\frac{1}{\sqrt{\beta_2 v_{t-1,i}}} - \frac{1}{\sqrt{\tilde{v}_t}}\right)m_{t-1,i} \tag{51}$$

$$\mathbb{E}[f(q_{t+1})]$$

$$\le f(q_t) + \mathbb{E}\langle\nabla f(q_t), q_{t+1} - q_t\rangle + \frac{L}{2}\mathbb{E}\|q_{t+1} - q_t\|^2$$

$$= f(q_t) - \eta\frac{1-\beta_1}{1-\frac{\beta_1}{\sqrt{\beta_2}}}\mathbb{E}\langle\nabla f(x_t), \frac{1}{\sqrt{\tilde{v}_t}}\odot\nabla f(w_t)\rangle - \eta\frac{1}{1-\frac{\beta_1}{\sqrt{\beta_2}}}\langle\nabla f(x_t), (\frac{1}{\sqrt{v_t}} - \frac{1}{\sqrt{\tilde{v}_t}})\odot m_t\rangle$$

$$+ \eta\frac{\beta_1}{1-\frac{\beta_1}{\sqrt{\beta_2}}}\langle\nabla f(x_t), (\frac{1}{\sqrt{\beta_2 v_{t-1}}} - \frac{1}{\sqrt{\tilde{v}_t}})\odot m_{t-1}\rangle + \mathbb{E}\langle\nabla f(q_t) - \nabla f(x_t), q_{t+1} - q_t\rangle$$

$$+ \frac{L}{2}\mathbb{E}\|q_{t+1} - q_t\|^2$$

Summing up the above inequality over $\{1, ..., T\}$ yields that

$$\mathbb{E}[f(q_{T+1})] - f(q_1)$$

$$\le -\frac{\eta(1-\beta_1)}{1-\frac{\beta_1}{\sqrt{\beta_2}}}\sum_{t=1}^{T}\sum_{l=1}^{d}\mathbb{E}\frac{\nabla f(x_t)_l\nabla f(w_t)_l}{\sqrt{\tilde{v}_{t,l}}} - \frac{\eta}{1-\frac{\beta_1}{\sqrt{\beta_2}}}\sum_{t=1}^{T}\sum_{l=1}^{d}\mathbb{E}\nabla f(x_t)_l m_{t,l}\left(\frac{1}{\sqrt{v_{t,l}}} - \frac{1}{\sqrt{\tilde{v}_{t,l}}}\right)$$

$$+ \frac{\eta\beta_1}{1-\frac{\beta_1}{\sqrt{\beta_2}}}\sum_{t=1}^{T}\sum_{l=1}^{d}\mathbb{E}\nabla f(x_t)_l m_{t-1,l}\left(\frac{1}{\sqrt{\beta_2 v_{t-1,l}}} - \frac{1}{\sqrt{\tilde{v}_{t,l}}}\right) + \frac{L}{2}\sum_{t=1}^{T}\mathbb{E}\|q_{t+1} - q_t\|^2$$

$$+ \sum_{t=1}^{T}\mathbb{E}\langle\nabla f(q_t) - \nabla f(x_t), q_{t+1} - q_t\rangle \tag{52}$$

Firstly, similar to (24), we obtain that

$$-\sum_{t=1}^{T}\sum_{l=1}^{d}\mathbb{E}\frac{\nabla f(x_t)_l\nabla f(w_t)_l}{\sqrt{\tilde{v}_{t,l}}}$$

$$\le -\frac{3}{4}\sum_{t=1}^{T}\sum_{l=1}^{d}\mathbb{E}\frac{\nabla f(x_t)_l^2}{\sqrt{\tilde{v}_{t,l}}} + \rho^2 L^2\sum_{t=1}^{T}\sum_{l=1}^{d}\mathbb{E}\frac{\frac{r_{t,l}^2}{u_{t,l}}}{\sqrt{\tilde{v}_{t,l}}}$$

$$\le -\frac{3}{4}\sum_{t=1}^{T}\sum_{l=1}^{d}\mathbb{E}\frac{\nabla f(x_t)_l^2}{\sqrt{\tilde{v}_{t,l}}} + \frac{(1-\beta_1)^2\rho^2 L^2}{(1-\frac{\beta_1}{\sqrt{\beta_2}})^2(1-\beta_2)^{3/2}\sqrt{D_0}}\sum_{t=1}^{T}\sum_{l=1}^{d}\mathbb{E}(\ln\frac{r_{T,i}}{\epsilon} - T\ln\beta_2)$$

$$\tag{53}$$

Secondly, following the derivation in Wang et al. (2023a), we have

$$\sum_{t=1}^{T}\sum_{l=1}^{d}\mathbb{E}\nabla f(x_t)_l m_{t,l}\left(\frac{1}{\sqrt{\tilde{v}_{t,l}}} - \frac{1}{\sqrt{v_{t,l}}}\right) \le \sum_{t=1}^{T}\sum_{l=1}^{d}\mathbb{E}|\nabla f(x_t)_l||m_{t,l}|\frac{(1-\beta_2)(g_{t,l}^2 + D_0)}{\sqrt{v_{t,l}}\sqrt{\tilde{v}_{t,l}}(\sqrt{v_{t,l}} + \sqrt{\tilde{v}_{t,l}})}$$

$$\tag{54}$$

For the above inequality

$$\sum_{t=1}^{T}\sum_{l=1}^{d}\mathbb{E}|\nabla f(x_t)_l||m_{t,l}|\frac{(1-\beta_2)g_{t,l}^2}{\sqrt{v_{t,l}}\sqrt{\tilde{v}_{t,l}}(\sqrt{v_{t,l}}+\sqrt{\tilde{v}_{t,l}})}\leq\frac{(1-\beta_1)\eta}{4(1-\frac{\beta_1}{\sqrt{\beta_2}})}\sum_{t=1}^{T}\sum_{l=1}^{d}\mathbb{E}\frac{\nabla f(x_t)_l^2}{\sqrt{\tilde{v}_{t,l}}}$$

$$\frac{2\eta\sqrt{(1-\beta_2)D_0}}{(1-\frac{\beta_1^2}{\beta_2})^2}\sum_{t=1}^{T}\sum_{l=1}^{d}\mathbb{E}\frac{g_{t,l}^2}{v_{t,l}}+\frac{4(1-\beta_1)\eta D_1}{(1-\frac{\beta_1}{\sqrt{\beta_2}})^2\sqrt{\beta_2}}\sum_{t=1}^{T}\sum_{l=1}^{d}\mathbb{E}(\frac{1}{\sqrt{\beta_2\tilde{v}_{t,l}}}-\frac{1}{\sqrt{v_{t,l}}})\nabla f(w_t)_l^2$$

$$(55)$$

Further, we have

$$\sum_{t=1}^{T}\sum_{l=1}^{d}\mathbb{E}\frac{\nabla f(w_t)_l^2}{\sqrt{\beta_2\tilde{v}_{t,l}}}$$

$$\leq\quad\sum_{t=1}^{T}\sum_{l=1}^{d}\mathbb{E}\frac{\nabla f(w_{t-1})_l^2}{\sqrt{\beta_2\tilde{v}_{t-1,l}}}+\frac{(1-\frac{\beta_1}{\sqrt{\beta_2}})(1-\beta_1)\sqrt{\beta_2}}{16D_1}\sum_{t=1}^{T}\sum_{l=1}^{d}\mathbb{E}\frac{\nabla f(w_t)_l^2}{\sqrt{\tilde{v}_{t,l}}}$$

$$+\frac{16(1+D_1)L^2}{\beta_2^{3/2}(1-\frac{\beta_1}{\sqrt{\beta_2}})(1-\beta_1)\sqrt{(1-\beta_2)D_0}}\sum_{t=1}^{T}\mathbb{E}\|w_t-w_{t-1}\|^2$$

$$\leq\quad\sum_{t=1}^{T}\sum_{l=1}^{d}\mathbb{E}\frac{\nabla f(w_{t-1})_l^2}{\sqrt{\beta_2\tilde{v}_{t-1,l}}}+\frac{(1-\frac{\beta_1}{\sqrt{\beta_2}})(1-\beta_1)\sqrt{\beta_2}}{16D_1}\sum_{t=1}^{T}\sum_{l=1}^{d}\mathbb{E}\frac{\nabla f(w_t)_l^2}{\sqrt{\tilde{v}_{t,l}}}$$

$$+\frac{16(1+D_1)L^2}{\beta_2^{3/2}(1-\frac{\beta_1}{\sqrt{\beta_2}})(1-\beta_1)\sqrt{(1-\beta_2)D_0}}\sum_{t=1}^{T}\sum_{l=1}^{d}\mathbb{E}(2\eta^2\frac{m_{t,l}^2}{v_t}+8\rho^2\frac{r_{t,l}^2}{u_{t,l}})\qquad(56)$$

Thus, we have

$$\sum_{t=1}^{T}\sum_{l=1}^{d}\mathbb{E}(\frac{1}{\sqrt{\beta_2\tilde{v}_{t,l}}}-\frac{1}{\sqrt{v_{t,l}}})\nabla f(w_t)_l^2\leq\sum_{t=1}^{T}\sum_{l=1}^{d}\mathbb{E}(\frac{\nabla f(w_{t-1})_l^2}{\sqrt{\beta_2\tilde{v}_{t,l}}}-\frac{\nabla f(w_t)_l^2}{\sqrt{v_{t,l}}})$$

$$+\frac{(1-\frac{\beta_1}{\sqrt{\beta_2}})(1-\beta_1)\sqrt{\beta_2}}{16D_1}\sum_{t=1}^{T}\sum_{l=1}^{d}\mathbb{E}\frac{\nabla f(w_t)_l^2}{\sqrt{\tilde{v}_{t,l}}}$$

$$+\frac{16(1+D_1)L^2}{\beta_2^{3/2}(1-\frac{\beta_1}{\sqrt{\beta_2}})(1-\beta_1)\sqrt{(1-\beta_2)D_0}}\sum_{t=1}^{T}\sum_{l=1}^{d}\mathbb{E}(2\eta^2\frac{m_{t,l}^2}{v_t}+8\rho^2\frac{r_{t,l}^2}{u_{t,l}})\qquad(57)$$

Substituting Lemma 9 into (57) yields that

$$\sum_{t=1}^{T}\sum_{l=1}^{d}\mathbb{E}(\frac{1}{\sqrt{\beta_2\tilde{v}_{t,l}}}-\frac{1}{\sqrt{v_{t,l}}})\nabla f(w_t)_l^2\leq\sum_{t=1}^{T}\sum_{l=1}^{d}\mathbb{E}(\frac{\nabla f(w_{t-1})_l^2}{\sqrt{\beta_2\tilde{v}_{t,l}}}-\frac{\nabla f(w_t)_l^2}{\sqrt{v_{t,l}}})$$

$$C_4\sum_{l=1}^{d}\mathbb{E}\ln r_{t,l}^2+C_5\sum_{l=1}^{d}\mathbb{E}\ln m_{t,l}^2+C_6\qquad(58)$$

Substituting (58) into (55) yields that

$$\sum_{t=1}^{T} \sum_{l=1}^{d} \mathbb{E}|\nabla f(x_t)_l||m_{t,l}| \frac{(1-\beta_2)g_{t,l}^2}{\sqrt{v_{t,l}}\sqrt{\tilde{v}_{t,l}}(\sqrt{v_{t,l}} + \sqrt{\tilde{v}_{t,l}})}$$

$$\leq \frac{(1-\beta_1)\eta}{4(1-\frac{\beta_1}{\sqrt{\beta_2}})} \sum_{t=1}^{T} \sum_{l=1}^{d} \mathbb{E} \frac{\nabla f(x_t)_l^2 + \nabla f(w_t)_l^2}{\sqrt{\tilde{v}_{t,l}}} + \frac{2\eta\sqrt{(1-\beta_2)D_0}}{(1-\frac{\beta_1^2}{\beta_2})^2} \sum_{t=1}^{T} \sum_{l=1}^{d} \mathbb{E} \frac{g_{t,l}^2}{v_{t,l}}$$

$$+ \frac{4(1-\beta_1)\eta D_1}{(1-\frac{\beta_1}{\sqrt{\beta_2}})^2} \sum_{t=1}^{T} \sum_{l=1}^{d} \mathbb{E}\left(\frac{\nabla f(w_{t-1})_l^2}{\sqrt{\beta_2 \tilde{v}_{t,l}}} - \frac{\nabla f(w_t)_l^2}{\sqrt{v_{t,l}}}\right)$$

$$+ C_4 \sum_{l=1}^{d} \mathbb{E}\ln r_{t,l}^2 + C_5 \sum_{l=1}^{d} \mathbb{E}\ln m_{t,l}^2 + C_6 \tag{59}$$

Then, we have

$$\sum_{t=1}^{T} \sum_{l=1}^{d} \mathbb{E}|\nabla f(x_t)_l||m_{t,l}| \frac{(1-\beta_2)D_0}{\sqrt{v_{t,l}}\sqrt{\tilde{v}_{t,l}}(\sqrt{v_{t,l}} + \sqrt{\tilde{v}_{t,l}})}$$

$$\leq \frac{(1-\beta_1)\eta}{8(1-\frac{\beta_1}{\sqrt{\beta_2}})} \sum_{t=1}^{T} \sum_{l=1}^{d} \mathbb{E} \frac{\nabla f(x_t)_l^2}{\sqrt{\tilde{v}_{t,l}}} + \frac{2\eta\sqrt{(1-\beta_2)D_0}}{(1-\beta_1)(1-\frac{\beta_1}{\sqrt{\beta_2}})} \sum_{t=1}^{T} \sum_{l=1}^{d} \mathbb{E} \frac{m_{t,l}^2}{v_{t,l}} \tag{60}$$

Substituting (59) and (60) into (54) yields that

$$\sum_{t=1}^{T} \sum_{l=1}^{d} \mathbb{E}\nabla f(x_t)_l m_{t,l}\left(\frac{1}{\sqrt{\tilde{v}_{t,l}}} - \frac{1}{\sqrt{v_{t,l}}}\right)$$

$$\leq \frac{3(1-\beta_1)\eta}{8(1-\frac{\beta_1}{\sqrt{\beta_2}})} \sum_{t=1}^{T} \sum_{l=1}^{d} \mathbb{E} \frac{\nabla f(x_t)_l^2 + \nabla f(w_t)_l^2}{\sqrt{\tilde{v}_{t,l}}} + \frac{2\eta\sqrt{(1-\beta_2)D_0}}{(1-\frac{\beta_1^2}{\beta_2})^2} \sum_{t=1}^{T} \sum_{l=1}^{d} \mathbb{E} \frac{g_{t,l}^2}{v_{t,l}}$$

$$+ \frac{4(1-\beta_1)\eta D_1}{(1-\frac{\beta_1}{\sqrt{\beta_2}})^2} \sum_{t=1}^{T} \sum_{l=1}^{d} \mathbb{E}\left(\frac{\nabla f(w_{t-1})_l^2}{\sqrt{\beta_2 \tilde{v}_{t,l}}} - \frac{\nabla f(w_t)_l^2}{\sqrt{v_{t,l}}}\right)$$

$$+ C_4 \sum_{l=1}^{d} \mathbb{E}\ln r_{t,l}^2 + C_5 \sum_{l=1}^{d} \mathbb{E}\ln m_{t,l}^2 + C_6 \tag{61}$$

Thirdly, similar to Wang et al. (2023a), we have

$$\sum_{t=1}^{T} \sum_{l=1}^{d} \mathbb{E}\nabla f(x_t)_l m_{t-1,l}\left(\frac{1}{\sqrt{\beta_2 v_{t-1,l}}} - \frac{1}{\sqrt{\tilde{v}_{t,l}}}\right)$$

$$\leq \frac{1}{16} \sum_{t=1}^{T} \sum_{l=1}^{d} \mathbb{E} \frac{\nabla f(x_t)_l^2}{\sqrt{\tilde{v}_{t,l}}} + \frac{4\beta_1\sqrt{1-\beta_2}\sqrt{D_0}}{(1-\beta_1)\beta_2} \sum_{t=1}^{T} \sum_{l=1}^{d} \mathbb{E} \frac{m_{t-1,l}^2}{v_{t,l}} \tag{62}$$

and

$$\sum_{t=1}^{T} \mathbb{E}\langle \nabla f(q_t) - \nabla f(x_t), q_{t+1} - q_t \rangle + \frac{L}{2}\mathbb{E}\|q_{t+1} - q_t\|^2$$

$$\leq \sum_{t=1}^{T} \eta^2 L\left(4\left(\frac{\frac{\beta_1}{\sqrt{\beta_2}}}{1-\frac{\beta_1}{\sqrt{\beta_2}}}\right)^2 \mathbb{E}\|\frac{1}{\sqrt{v_{t-1}}} \odot m_{t-1}\|^2 + 3\left(\frac{1}{1-\frac{\beta_1}{\sqrt{\beta_2}}}\right)^2 \mathbb{E}\|\frac{1}{\sqrt{v_t}} \odot m_t\|^2\right) \tag{63}$$

$$\square$$

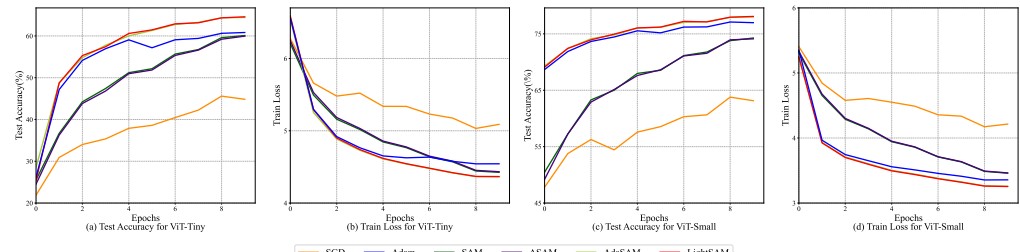

Figure 1: Experimental results of fine-tuning ViT models on Imagenet. (a): Test accuracy w.r.t. epochs for ViT-Tiny; (b): Train loss w.r.t. epochs for ViT-Tiny; (c): Test accuracy w.r.t. epochs for ViT-Small; (d) Train loss w.r.t. epochs for ViT-Small.

Next, substituting (53), (61), (62) and (63) into (52) and bounding the term $\sum_{t=1}^{T}\sum_{l=1}^{d}\frac{g_{t,l}^2}{v_{t,l}}$, $\sum_{t=1}^{T}\sum_{l=1}^{d}\frac{m_{t,l}^2}{v_{t,l}}$, $\sum_{t=1}^{T}\sum_{l=1}^{d}\frac{r_{t,l}^2}{u_{t,l}}$ with Lemma 5,10 yields that

$$\sum_{t=1}^{T}\sum_{l=1}^{d}\mathbb{E}\frac{\nabla f(x_t)_l^2}{\sqrt{\tilde{v}_{t,l}}} \leq C_7 + C_8\sum_{l=1}^{d}\mathbb{E}\ln u_{t,l}, \tag{64}$$

where $C_7$ and $C_8$ are constants with respect to $\eta$, $\rho$, $\beta_1$, $\beta_2$, $D_0$ and $D_1$. Finally, following Wang et al. (2023a) to bound $\mathbb{E}\sqrt{\tilde{v}_{t,l}}$ and the final steps in proof of Theorem 1, we obtain the $O(\ln T/\sqrt{T})$ convergence rate.

## B  EXPERIMENT ILLUSTRATION

We plot the curves of training loss and test accuracy of fine-tuning ViT models in Figure 1. From the figure, we could observe that regardless of the test accuracy and training loss, AdaSAM and our proposed algorithm LightSAM are ahead of other baselines obviously throughout the whole process, and LightSAM has a little advantage over AdaSAM. Though this performance is partly due to the power of Adam in Transformer-based model, it still illustrates the capability of adopting adaptive hyper-parameters in the SAM optimizer.

