# OpenReview forum: "LightSAM: Parameter-Agnostic Sharpness-Aware Minimization"
_ICLR.cc/2025/Conference — ICLR 2025 Conference Withdrawn Submission_

### Official Review · Reviewer_bVHw · 2024-10-24

**Soundness:** 3
**Presentation:** 2
**Contribution:** 2
**Rating:** 6
**Confidence:** 4

**Summary:**

This paper proposes LightSAM, a new variant of the Sharpness-Aware Minimization (SAM) optimizer that incorporates three adaptive optimization techniques (AdaGrad-Norm, AdaGrad, and Adam). The key innovation is using these adaptive methods for both the perturbation radius and learning rate in SAM, eliminating the need for parameter tuning. The main theoretical contribution is proving $O(log T/\sqrt{T})$ convergence for LightSAM under weaker assumptions than prior work, notably removing the gradient boundedness requirement and achieving parameter-agnostic properties. Empirically, the authors evaluate LightSAM on image classification tasks using MNIST and ImageNet datasets. The experiments demonstrate that LightSAM achieves comparable or better performance than carefully tuned SAM while being insensitive to parameter choices, validating its parameter-agnostic properties.

**Strengths:**

- Algorithm that makes SAM parameter-agnostic by removing hyperparameter restrictions on convergence
- Provides theoretical analysis with $O(\log T/ \sqrt{T})$ convergence guarantees under weaker assumptions
- Demonstrates consistent improvements over AdaSAM across different datasets

**Weaknesses:**

- My main issue is the mismatch between theory and empirics: the theoretical results focus on optimization convergence for differentiable loss landscapes, while the empirical results demonstrate generalization performance. This disconnect makes it challenging to assess the practical relevance of the theoretical contribution.

- Limited discussion of computational overhead, especially the extra overhead that comes from accumulating historical gradients, how long does it take in the experiments? Any quantitative comparison with baseline methods?

- The convergence guarantee doesn't offer new insights into generalization properties. (At least doesn't explain it well in the paper.)

**Questions:**

- Why the analysis is free to the parameter $\rho$? In Theorem 1, we can see that $A_1$ is related to the term $\eta^{2} -2\rho^{2}\log(\epsilon)$ which would be negative if ρ is too large. Better add some explanation.

- While the perturbation radius $\rho$ does not affect the convergence result, how does it influence the model's generalization ability in this paper? Any empirical or theoretical insights provided?

- In line 228, the authors mentioned "Technical Challenge", but what's the key technique to address this challenge? This needs to be explained.

- In Section 3.3, the authors provide the analysis of LightSAM-II (AdaGrad), but without explanation. How to understand these theorems? What insights do they provide?

---

> ### Author Response · Authors · 2024-11-26
>
> >The mismatch between theory and empirics: the theoretical results focus on optimization convergence for differentiable loss landscapes, while the empirical results demonstrate generalization performance.
>
> The goal of [1] is to find a minima with flat loss landscape, thereby increasing the generalization ablibity of the model. By forming the above goal as a minimax optimization problem and solving it, [1] proposes the SAM optimizer with an extra perturbation step. Since we aim to make SAM parameter-agnostic, we present the generalization performance in the experiments. This organization is adopted other SAM-related works. The convergence performances are listed in section "B. Experiment Illustration" in the Appendix.
>
> [1] Foret P, Kleiner A, Mobahi H, et al. Sharpness-aware minimization for efficiently improving generalization[J]. arXiv preprint arXiv:2010.01412, 2020.
>
> >In Theorem 1, we can see that $A_1$ is related to the term $\eta-2\rho^2 \log(\epsilon)$ which would be negative if $\rho$ is too large.
>
> In fact, $\eta-2\rho^2 \log(\epsilon)$ would always be positive since $\epsilon$ is a real number less than 1 (commonly set to $10^{-3}$ or $10^{-4}$ in the experiments).
>
> >While the perturbation radius $\rho$ does not affect the convergence result, how does it influence the model's generalization ability in this paper? Any empirical or theoretical insights provided?
>
> We vary the value of $\rho$ to evaluate the stability of LightSAM, and the results are listed in Table 3,5 and global comment above. We could observe that with different values of $\rho$, LightSAM always performs well and achieves the lowest standard deviation.
>
> >In line 228, the authors mentioned "Technical Challenge", but what's the key technique to address this challenge?
>
> We could summarize two points of "Technical Challenge", the first is to decouple the randomness in the numerator and denominator of one term and the second is to bound the terms concerning both $\mathbb{E}||\nabla f(x_t)||^2$ and $\mathbb{E}||\nabla f(w_t)||^2$. To solve the first challenge, we turn to bound term $T_2$ in the proof of Theorem 4 (in the Appendix). The term $T_2$ concerning $\nabla f(x_t)$ and $\nabla f(w_t)$, thus corresponding to the second challenge. To solve it, we turn to bound $\sum ||\nabla f(w_t)||^2(\frac{1}{\sqrt{v_{t-1}}}-\frac{1}{\sqrt{v_t}})$ as shown in equation (23) and propose Lemmas 6 and 7 to bound it. Finally, Lemma 8 establishes the inequality relationship between $\nabla f(x_t,\xi_t)$ and $\nabla f(w_t,\xi_t)$ and finishes the proof.
>
> >In Section 3.3, the authors provide the analysis of LightSAM-II (AdaGrad), but without explanation.
>
> The difference between LightSAM-II and LightSAM-I is that the multiplication and division becomes element-wise for vectors, thus, the forms and explanations of the theoretical results for these two approaches are similar. Thus, we omit the further explanation of LightSAM-II to prevent the duplication.

---

> > ### Comment · Reviewer_bVHw · 2024-11-27
> >
> > Thank you for your response, I appreciate the clarification on theoretical aspects. After reviewing the explanations, I maintain my score of weak accept. A few areas could be strengthened: computational overhead analysis would help evaluate practical efficiency, the notation $-\log(\epsilon)$ could be clearer as $\log(1/\epsilon)$, and the LightSAM-II analysis would benefit from more detailed explanation given its unique element-wise operations. Nevertheless, the paper makes contributions to SAM.

---

> > > ### Author Response · Authors · 2024-11-29
> > > **Response to Reviewer bVHw**
> > >
> > > Dear reviewer bVHw, thank you very much for your affirmation of this work. We would keep strengthening our work.

---

### Official Review · Reviewer_Sp3H · 2024-10-29

**Soundness:** 1
**Presentation:** 2
**Contribution:** 2
**Rating:** 3
**Confidence:** 4

**Summary:**

The paper presents LightSAM, a parameter-agnostic variant of the Sharpness-Aware Minimization (SAM) optimizer, designed to enhance generalization by reducing sensitivity to hyperparameters.
LightSAM replaces the SGD optimizer in SAM with adaptive optimizers (AdaGrad-Norm, AdaGrad, and Adam) for both weight perturbation and model updating steps, aiming to achieve convergence without the need for parameter tuning.
The theoretical analysis suggests that LightSAM converges with a rate of $\mathcal{O}\left(\ln T/\sqrt{T}\right)$ under relaxed assumptions, and empirical evaluations on MNIST and ImageNet support its effectiveness.

**Strengths:**

- LightSAM effectively addresses a main limitation of SAM by making it parameter-agnostic, potentially simplifying hyperparameter tuning in practical applications.
- The empirical results are promising. LightSAM’s performance on MNIST and ImageNet, particularly its insensitivity to hyperparameters, demonstrates its practical potential and competitive accuracy.

**Weaknesses:**

I have a major concern regarding the soundness of the proof. I think that the authors ignore the correlation of $w_t$ and $\xi_t$. Then, two problems arise. First, due to this correlation, the equality such as Line 836-837 does not hold. Second, when applying $w_t$ on affine variance noise, the authors use a form like $\mathbb{E}\\|\nabla f(w\_t,\xi\_t)\\|^2 \le D_0+D_1 \mathbb{E}\\|\nabla f(w\_t)\\|^2$ which is not the form of Assumption 2 as there is no expectation on RHS of Assumption 2. I think that $\\|\nabla f(w\_t,\xi\_t)\\|$ could not be directly bounded through Assumption 2.

Besides, I think that it's better to use notation like $\mathbb{E}_t$ to denote the conditional expectation. It's confusing for readers to see $\mathbb{E}$ denotes both the conditional expectation and the full expectation

I have another major concern regarding the motivation of the algorithm. The main motivation in this paper comes from incorporating the adaptive perturbation radius to eliminate the tuned parameter. In comparison to the SAM, the major difference comes from using $\\|u\_t\\|$, the cumulative sum of the gradient norm instead of the gradient norm $\\|g_t\\|$ in the perturbation radius. However, I think that both two ways could be regarded as adaptive perturbation radiuses since the radius $\rho\_1 = \rho/ \\|u\_t\\| $ and $\rho\_2 = \rho/\\|g_t\\|$ are all determined by the gradient information. So I am not sure whether it's necessary to incorporate the $\\|u\_t\\|$, or are there any essential differences between the two $\rho$? The author could claim more on this, maybe showing that the original SAM must tune the perturbation radius unless diverging.

There are some other concerns as follows.

- The proof methodology in this paper bears a notable resemblance to that in the referenced AdaGrad convergence paper [Wang et al., 2023].
	A more in-depth discussion on how the theoretical contributions in this work differ fundamentally from existing proofs could better clarify the unique theoretical foundation of LightSAM and highlight its distinct contributions.
- The paper could benefit from a comparative discussion on computational complexity between LightSAM and other SAM-related optimizers. As computational overhead is a key consideration for optimizers, especially in large-scale models, providing this comparison would give a more complete view of LightSAM’s practical costs.
- While MNIST and ImageNet are standard datasets, they may not fully demonstrate LightSAM’s potential across various complex, real-world tasks.
	Including additional experiments on tasks beyond classification, such as natural language processing or reinforcement learning, could provide a more comprehensive evaluation.

**Questions:**

- There are some typographical errors in the proofs that affect readability and accuracy. For instance, in Equation (22) of the proof for Theorem 4, $\frac{1}{\sqrt{v_{t-1}}}$ is placed outside the summation symbol, which might lead to misunderstandings.
	Addressing these would enhance the overall precision and presentation quality of the paper.
- The uniform size of parentheses throughout the proofs makes it challenging to follow nested expressions, which affects readability.
	Adjusting parentheses sizes to reflect the level of nesting could improve clarity and help readers follow the steps more comfortably.

---

> ### Author Response · Authors · 2024-11-26
>
> >The inequality such as Line 836-837 does not hold. $||\nabla f(w_t,\xi_t)||$ could not be directly bounded through Assumption 2.
>
> Firstly, for the equality at Line 836-837, since the term $||\nabla f(w_t)||$ and $v_{t-1}$ do not have randomness at time $t$, we pull them out of the expectation. Secondly, we could obtain that $\mathbb{E}||\nabla f(w_t,\xi_t)||^2 \leq D_0 + D_1||\nabla f(w_t)||^2$ by Assumption 2, here the expectation operation removes the randomness caused by the stochastic sampling $\xi_t$. In other words, this expectation is taken conditioned on the historical stochastic gradients. Thus, when we take global expectation on the LHS of equation (13), we are actually taking expectation on a conditioned expectation, i.e. $\mathbb{E}[\mathbb{E}[a|b]] = \mathbb{E}[a]$, thereby we finally add the expectation notation on $||\nabla f(w_t)||^2$.
>
> >Both two ways could be regarded as adaptive perturbation radius since $\rho_1=\rho/||u_t||$ and $\rho_2=\rho/||g_t||$ are all determined by the gradient information. Whether it's necessary to incorporate the $||u_t||$ or are there any essential differences?
>
> Firstly, the perturbation radius adopted in original SAM $\rho_2=\rho/||g_t||$ uses the gradient normalization (discussed in [1]), not the adaptive perturbation radius. Our adopted perturbation radius $\rho_1=\rho/||g_t||$ is adaptive, the same as previous adaptive optimizers (Adagrad-Norm, Adagrad and Adam). Secondly, AdaSAM [2] uses the perturbation radius $\rho/||g_t||$ and the adaptive learning rate. However, AdaSAM is not proved to be parameter-agnostic as our approach. Thus, we think it is necessary to incorporate $||u_t||$ to make SAM parameter-agnostic.
>
> >The theoretical contributions in this work differ fundamentally from existing proofs.
>
> Our approach involves two gradients $\nabla f(x_t)$ and $\nabla f(w_t)$, thus the proof is more complex than [3]. For example, the proof of [3] needs to bound the term $\mathbb{E}||\nabla f(x_t)||^2(\frac{1}{\sqrt{v_{t-1}}}-\frac{1}{\sqrt{v_t}})$, while we need to bound the term $\mathbb{E}||\nabla f(w_t)||^2(\frac{1}{\sqrt{v_{t-1}}}-\frac{1}{\sqrt{v_t}})$. As a consequence, we propose Lemmas 1 and 2 to bound it. The analysis in Lemmas 1 and 2 is non-trivial, please refer to the Appendix for more details.
>
> > Including additional experiments on tasks beyond classification, such as natural language processing or reinforcement learning, could provide a more comprehensive evaluation.
>
> Thanks for your suggestion. We supplement the experiment on GLUE benchmark and list the results in the global comment above, please refer to it.
>
> >There are some typographical errors in the proofs that affect readability and accuracy.
>
> Thanks for your comment. We have corrected the error.
>
> [1] Dai, Yan, Kwangjun Ahn, and Suvrit Sra. "The crucial role of normalization in sharpness-aware minimization." Advances in Neural Information Processing Systems 36 (2024).
>
> [2] Sun, Hao, et al. "Adasam: Boosting sharpness-aware minimization with adaptive learning rate and momentum for training deep neural networks." Neural Networks 169 (2024): 506-519.
>
> [3] Wang, Bohan, et al. "Convergence of adagrad for non-convex objectives: Simple proofs and relaxed assumptions." The Thirty Sixth Annual Conference on Learning Theory. PMLR, 2023.

---

> > ### Comment · Reviewer_Sp3H · 2024-11-28
> >
> > I do not think that $\nabla f(w_t)$ does not have randomness at time $t$ since $w_t$ is determined by $\xi_t$ from e.g.,  Algorithm 1 (Line 5) as well as Algorithm 2 and 3. Given this, if you just take the conditional expectation on $\\|\nabla f(w_t,\xi_t)\\|$, the RHS should be independent of the randomness at time $t$, specifically $\xi_t$. However, there arises $w_t$ in RHS, still correlating with $\xi_t$.
> >
> > I take two specific examples in your proof:
> >
> > - Line 836-837, I will regard $\mathbb{E}$ as the conditional expectation here. Then, since $\nabla f(w_t)$ is correlated with $\xi_t$, it can not be pulled out.
> >
> > - Line 839-840. There is no expectation notation before $\\|\nabla f(w_t,\xi_t)\\|/v_{t-1}^{1/4}$ in Line 837. Why there arise an additional expectation notation in Line 839 by only using Young's inequality?
> >
> > I hope the authors can fully address these concerns.

---

### Official Review · Reviewer_FjrZ · 2024-11-02

**Soundness:** 3
**Presentation:** 3
**Contribution:** 2
**Rating:** 6
**Confidence:** 4

**Summary:**

This work studies the SAM algorithm with  3 adaptive optimizers and provide theoretical analyses, showing its agnostic properties on the convergence results w.r.t. the perturbation. The perspective is interesting, but lacks sufficient explanations and numerical evidence. I would like to raise my scores, if my concerns are thoroughly addressed both analytically and numerically, preferably.

**Strengths:**

The convergence results being agnostic to parameter values, i.e., the perturbation radius, are interesting to the field.

**Weaknesses:**

The benefits of the so-called Light SAM are not sufficiently presented in the experiments. The reasons of parameter-agnostic property in convergences are lacking, so are the extension to other optimizers aside of the presented 3.

**Questions:**

Questions & some advice:

1. The main contribution of this work to me lies in the parameter-agnostic convergence results of SAM adopting to the adaptive optimizers presented in this work. However, in the experiments, such benefits are not elaborated. Firstly, the shown results only appear with incremental and quite marginal improvement; secondly, why not vary some parameters, e.g., $\rho$, to compare the results on acc and especially on convergence as claimed by the theorems; thirdly, more ablation studies should done for each of the proposed 3 versions.  E.g.,

- can it be understood that the proposed method only gives marginal improvements in table 4 and 5?

- the convergence plots for a range of $\rho$ or $\eta$ values can be helpful to provide a more comprehensive ablation study;

- compare the three proposed versions with other optimizers across different hyperparameter settings, rather than a single setup of hyperparameters in table 5; otherwise, how the hyperparameters are determined in comparisons.

2. Can other optimizers also get such parameter agnostic properties, aside of the presented 3? What’s essential characteristics of those optimizers contributing to such agnostic properties in convergence?

-  discuss the key properties of the three optimizers that enable parameter-agnostic convergence, and whether these properties might extend to other optimizers.

3. Some minor aspects:  (a) why name it as LightSAM, as there are more efficient SAM-related algorithms [1][2] (not limited to). Please justify the choice of name.  (b) I would suggest specify the footnote 1 in a earlier place, e.g., the abstract, or maybe reconsider the title with more focus on the convergence perspective, because the agnostic property lies in the convergence analysis rather than the hyperparameter tuning, if possible.

 [1] Revisiting Random Weight Perturbation for Efficiently Improving Generalization, TMLR, 2024.

[2] Friendly sharpness-aware minimization, CVPR, 2024.

---

> ### Author Response · Authors · 2024-11-26
>
> >Can it be understood that the proposed method only gives marginal improvements in table 4 and 5?
>
> Yes, the improvements in the experiments of this work is marginal. However, our goal is to propose a SAM-type algorithm that achieves the parameter-agnostic property, as mentioned on the top of page 2 "Can we make SAM parameter-agnostic?", not to propose an approach that outperforms previous algorithms in the experiments. The theoretical and experimental results have verified that we realize this goal.
>
> >The convergence plots for a range of $\rho$ or $\eta$ values can be helpful to provide a more comprehensive ablation study. Compare the three proposed versions with other optimizers across different hyperparameter settings, rather than a single setup of hyperparameters in table 5.
>
> In fact, we have already evaluated our proposed algorithm across different hyper-parameter settings. In Table 3, we adopt four hyper-parameter settings ($\rho$,2$\rho$)\*($\eta$,2$\eta$), where $\rho$ and $\eta$ are the values adopted to lead to results in Table 2. In Table 5, the nine results for one baselines are obtained under nine different hyper-parameter settings (0.5$\rho$,$\rho$,2$\rho$)\*(0.5$\eta$,$\eta$,2$\eta$), where $\rho$ and $\eta$ are the values adopted to lead to results in Table 4. A clearer presentation could be referred to the Table in the global comment above. We have mentioned these points in the paragraph "Sensitivity to hyper-parameters".
>
> >Can other optimizers also get such parameter agnostic properties, aside of the presented 3? What’s essential characteristics of those optimizers contributing to such agnostic properties in convergence?
>
> We have listed some of the existed parameter-agnostic optimizers in the section "Related Work". Among them, the techniques of Normalized SGDM and adaptive optimizers (Adagrad-Norm, Adagrad and Adam) are classical. Since the technique of adaptive learning rate is similar to normalization (upper bounding the gradient norm), we think the gradient normalization or bounding the weight update vector may be essential for the agnostic property.
>
> >Why name is as LightSAM?
>
> LightSAM is proved to converge under any hyper-parameter settings, therby reducing the heavy workload of tuning parameters. Thus, we use "light" to name it.

---

> > ### Comment · Reviewer_FjrZ · 2024-11-27
> > **Response to the rebuttal**
> >
> > Thanks for the responses. Most of the raised comments have been replied, while the lack of more elucidation on the convergence performances remained, as this is one of the key elements in the contribution claimed by the theorems. Despite that the appendix gives some examples in this aspect, more ablations and analyses on the convergence are still appreciated in the main body which mainly presents the generalization performances.
> >
> > Based on the current version I will maintain the evaluation scores w.r.t. 3 aspects of soundness, presentation and contribution. Since I  would acknowledge the enhanced understandings towards SAM in convergence with opinion of weak acceptance, the rating is raised to 6.

---

> > > ### Author Response · Authors · 2024-11-29
> > > **Response to Reviewer FjrZ**
> > >
> > > Dear reviewer FjrZ, thank you very much for your affirmation of this work.

---

### Official Review · Reviewer_xk1b · 2024-11-03

**Soundness:** 2
**Presentation:** 3
**Contribution:** 2
**Rating:** 5
**Confidence:** 4

**Summary:**

The paper introduces a variant of the Sharpness-Aware Minimization (SAM) optimizer designed to adjust both the learning rate and perturbation radius adaptively for SAM. The proposed method, LightSAM, addresses this by integrating adaptive optimizers, including AdaGrad-Norm, AdaGrad, and Adam, to automatically adjust these parameters without problem-dependent tuning, making it parameter-agnostic. The authors provide theoretical insights demonstrating LightSAM’s convergence rate, $O(\ln T/ \sqrt{T})$.  This paper also provides empirical validation through experiments on MNIST and Imagenet datasets, using LeNet and Vision Transformers. The results show that LightSAM achieves competitive performance to existing SAM-based methods, while reducing the dependency on hyper-parameter tuning.

**Strengths:**

- The paper develops a theoretical analysis of LightSAM’s convergence properties under adaptive learning rates and minimal assumptions, showing that it achieves a convergence rate of $O(\ln T/ \sqrt{T})$

- The experiments on diverse datasets, including MNIST and Imagenet, demonstrate LightSAM’s robustness and parameter-agnostic nature.

- The paper includes comprehensive comparisons with SAM, AdaSAM, and other optimizers, showing that LightSAM achieves comparable accuracy while reducing tuning complexity.

**Weaknesses:**

It is unclear how the hyper-parameter tuning complexity should be evaluated in the setting. While the paper has shown that under nine hyper-parameter settings, the LightSAM's standard deviation is small, it would be better to explain the protocol. Moreover, It would be better to conduct thorough studies on the hyper-parameter tuning (for example, under a larger range of hyper-parameters, the results are similar), and more experimental settings (for other types of tasks).

**Questions:**

- From the experiments, the proposed algorithm performs on par with the AdaSam baseline, while the AdaSam baseline is claimed to have a better convergence rate. Can the authors extend the discussion with AdaSam and explain the advantages of the proposed method?

- How should one decide the initial parameter of the LightSAM?

---

> ### Author Response · Authors · 2024-11-26
>
> >From the experiments, the proposed algorithm performs on par with the AdaSam baseline, while the AdaSam baseline is claimed to have a better convergence rate.
>
> The goal of this work is to propose a SAM-type algorithm with adaptive hyper-parameters and verify the parameter-agnostic property of it, not to propose an approach that outperforms previous algorithms in the experiments. The theoretical analysis and experimental results have already veried that LightSAM is parameter-agnostic. The theoretical contribution of this work is to prove the convergence rate of LightSAM under nearly the weakest assumptions, and do not obtain a better convergence rate than previous algorithms. Since LightSAM and AdaSAM both adopt the adaptive learning rate in the gradient descent step, it is trivial that they achieve similar performances.
>
> >How should one decide the initial parameter of the LightSAM?
>
> Since LightSAM adopts the adaptive learning rate in the gradient descent step, we set $\eta$ the same as the value commonly used by Adam optimizer, such as $1e-4$ in ImageNet experiment and $1e-5$ in GLUE experiment (these values are suggested in the experiments in previous studies). Then, the weight perturbation step adopts the same type of hyper-parameter as the gradient descent step, thus we set $\rho$ the same as $\eta$.

---

> > ### Comment · Reviewer_xk1b · 2024-11-27
> >
> > Thanks to the authors for their responses!
> >
> > Please also respond to my question in the weakness section: "It is unclear how the hyper-parameter tuning complexity should be evaluated in the setting."

---

> > > ### Author Response · Authors · 2024-11-29
> > > **Response to Reviewer xk1b**
> > >
> > > Dear reviewer xk1b:
> > >
> > > In fact, to our knowledge, there is a lack of a commonly accepted metric to evaluate the hyper-parameter tuning complexity. Most machine learning studies tune the hyper-parameters of the optimizers with the grid search, such as choosing the learning rate in the set {0.01,0.02,0.05,0.1,0.2,...} by comparing the performances. The complexity is decided by the size of the set. By thinking in reverse, we propose to adopt the same size of set or parameter-tuning method for each baseline (such as adopting nine hyper-parameter settings $(0.5\rho, \rho, 2\rho)$\*$(0.5\eta, \eta, 2\eta)$ in Table 5 in our work), then, the algorithm with more stable performances is more convenient and friendly for parameter-tuning.  Thus, in our work, we test the performances under nine hyper-parameter settings for each algorithm and compare their mean and standard deviation performances.

---

### Author Response · Authors · 2024-11-26
**Supplementary experiments on different type of dataset**

Great thanks for all the efforts from the reviewers.

We enrich the experiment part by supplementing the fine-tuning experiment on the GLUE benckmark which consists of eight tasks. We count the Matthew's correlation for CoLA, Pearson correlation for STS-B, F1 score for MRPC, averaged accuracy for MNLI, and accuracy for other tasks. The results are listed as below.

| Algorithms | CoLA | STS-B | MRPC | RTE | SST2 | MNLI | QNLI | QQP |
|  :----:   |  :----:   |  :----:   | :----:   |  :----:   |  :----:   |  :----:   |  :----:   | :----:   |
| SGD | 59.39 | 87.85 | 91.65 | 76.53 | 93.69 | 86.33 | 89.27 | 91.49 | 84.53 |
| Adam | 62.08 | 90.77 | 92.50 | 78.70 | 94.84 | 87.42 | 92.82 | 91.90 | 86.38 |
| SAM | 61.71 | 89.25 | 92.01 | 79.42 | 94.27 | 86.42 | 89.53 | 91.38 | 85.50 |
| ASAM | 63.51 | 89.14 | 92.48 | 78.70 | 93.81 | 86.44 | 90.17 | 91.57 | 85.73 |
| AdaSAM | 62.11 | 90.55 | 93.12 | 80.14 | 95.30 | 87.57 | 93.10 | 92.01 | 86.74 |
| LightSAM | 63.77 | 90.77 | 93.33 | 81.95 | 95.41 | 87.63 | 92.92 | 92.04 | 87.23 |

We also evaluate the stability of algorithms by test their performances on the STS-B task under nine sets of hyper-parameters. The method to set parameters are the same as ViT experiment. The results are listed as below.

SAM:
|Hyper-parameters $(\eta, \rho)$| (5e-3,2.5e-3) | (5e-3,5e-3) | (5e-3,0.01) | (0.01,2.5e-3) | (0.01,5e-3) | (0.01,0.01) | (0.02,2.5e-3) | (0.02,5e-3) | (0.02,0.01) |
|  :----:   |  :----:   |  :----:   | :----:   |  :----:   |  :----:   |  :----:   |  :----:   | :----:   |  :----:   |
|Accuracy| - | $89.53$ | $87.87$ | $89.31$ | $89.25$ | $89.19$ | - | - | - |

ASAM:
|Hyper-parameters $(\eta, \rho)$| (5e-3,5e-3) | (5e-3,0.01) | (5e-3,0.02) | (0.01,5e-3) | (0.01,0.01) | (0.01,0.02) | (0.02,5e-3) | (0.02,0.01) | (0.02,0.02) |
|  :----:   |  :----:   |  :----:   | :----:   |  :----:   |  :----:   |  :----:   |  :----:   | :----:   |  :----:   |
|Accuracy| $85.74$ | $83.26$ | - | $88.99$ | $89.14$ | $88.58$ | - | - | - |

AdaSAM:
|Hyper-parameters $(\eta, \rho)$| (5e-6,5e-3) | (5e-6,0.01) | (5e-6,0.02) | (1e-5,5e-3) | (1e-5,0.01) | (1e-5,0.02) | (2e-5,5e-3) | (2e-5,0.01) | (2e-5,0.02) |
|  :----:   |  :----:   |  :----:   | :----:   |  :----:   |  :----:   |  :----:   |  :----:   | :----:   |  :----:   |
|Accuracy| $90.20$ | $90.29$ | $90.27$ | $90.54$ | $90.55$ | $90.48$ | $90.86$ | $91.01$ | $90.92$ |

LightSAM:
|Hyper-parameters $(\eta, \rho)$| (5e-6,5e-6) | (5e-6,1e-5) | (5e-6,2e-5) | (1e-5,5e-6) | (1e-5,1e-5) | (1e-5,2e-5) | (2e-5,5e-6) | (2e-5,1e-5) | (2e-5,2e-5) |
|  :----:   |  :----:   |  :----:   | :----:   |  :----:   |  :----:   |  :----:   |  :----:   | :----:   |  :----:   |
|Accuracy| $90.42$ | $90.31$ | $90.39$ | $90.79$ | $90.77$ | $90.69$ | $90.97$ | $91.09$ | $91.05$ |

Average accuracy:
|Algorithms|SAM|ASAM|AdaSAM|LightSAM|
|  :----:   |  :----:   |  :----:   | :----:   |  :----:   |
|Accuracy| $88.97 \pm 0.79$ | $87.14\pm2.57$ |  $90.57 \pm 0.30$ |  $90.72 \pm 0.29$ |

From these results, we could observe that our proposed algorithm achieves the best performances in most tasks of GLUE benchmark. In addition, LightSAM also has the highest average accuracy and lowest standard deviation, indicating its insensitivity to hyper-parameters.

---

### Note · Authors · 2024-12-06

I have read and agree with the venue's withdrawal policy on behalf of myself and my co-authors.